# Effect of Surface Polymeric Organosilicon Nanolayers on the Electrochemical and Corrosion Behavior of Copper

**DOI:** 10.3390/polym16213066

**Published:** 2024-10-31

**Authors:** M. A. Petrunin, T. A. Yurasova, A. A. Rybkina, L. B. Maksaeva

**Affiliations:** Frumkin Institute of Physical Chemistry and Electrochemistry, Russian Academy of Sciences, Moscow 119071, Russia; tatal111@yandex.ru (T.A.Y.); aa_rybkina@mail.ru (A.A.R.); lmaksaeva@mail.ru (L.B.M.)

**Keywords:** metal corrosion, copper, organosilanes, self-assembled polymeric siloxane surface nanolayers, corrosion inhibition

## Abstract

The formation of polymeric self-organizing organosilicon surface nanolayers on copper occuring as a result of modification of the metal surface with organosilane-based formulations has been studied. The anticorrosive effect of such surface layers in corrosive chloride-containing electrolytes as well as in artificial and natural atmospheres has been studied in detail. It has been found that the maximum protective effect is observed at a thickness of 3.8 molecular layers, where the densest cross-linked polymer layers are formed that hinder the adsorption of chloride ions and other corrosive agents on the metal surface, thus significantly reducing the rate of their reactions with the surface copper atoms, and, as a result, inhibiting the corrosion and local anodic dissolution of the metal.

## 1. Introduction

Copper and its alloys have been important structural materials since ancient times. It is believed that the earliest discovered handmade metal object is a perforated copper pendant that was found in northern Iraq and is dated to 9500 BC [1]. The works of ancient authors, in particular, Pliny the Elder [1], Herodotus [2], and Olympiodorus [1,3], mention copper, bronze, brass, and items made from these materials. It is not coincidental that one of the key eras of human development (intermediate between the Neolithic and Bronze Age, in the IV-III millennia BC) was named the Copper Age or Copper Stone Age (Chalcolithic or Eneolithic). Moreover, it is believed [2] that the discovery and widespread use of copper brought the Stone Age to an end, providing much more efficient capabilities in the manufacture of life-essential tools, weapons, and other items [4] than were previously made from materials such as sharpened flint, bone, or wood. Copper is the 23rd most abundant element in the Earth’s crust, making up 0.006% of the crust [5].

Copper occupies a special place among metals because of its high electrical and thermal conductivity [5,6]. Copper is widely used in electrical engineering (for making cables and the conductive parts of electrical units), microelectronics, water supply, construction [6,7], motor vehicles [5], as a material of heat-sink tubes in heat exchangers, and in objects of cultural and historical heritage. Ancient philosophers and natural scientists attributed copper to the seven basic metals [8]. Copper is the main component of brasses, bronzes, copper-nickel, and other alloys that possess a number of valuable properties such as good electrical conductivity, ductility, and sufficiently high strength [6]. In addition, copper catalysts are used in some organic syntheses (hydrogenation of fatty acids, oxidation of propylene to acrolein, etc.) [6]. The widespread industrial use of copper is responsible for the huge volumes of copper mining and production. In fact, by the end of the 20th century, copper was in second place (after aluminum) in terms of industrial production volume [6,7]. However, copper, like most metals, is subject to corrosion [9], the damage from which can be significant. For example, the destruction of copper and bronze historical monuments may result not only in significant costs for their restoration [10,11,12] but also in irreparable losses for the world cultural community. The corrosion of copper and its alloys becomes especially important in the presence of corrosive anions (chlorides) in water and in the atmosphere which cause accelerated corrosion damage [9,10] of production systems, such as heat exchange, desalination systems, etc. In addition, special attention has been paid in recent years to environmental safety, since corrosion is accompanied by leaching of toxic copper ions [13,14] into the environment [15] that may harmfully affect the environmental situation [16].

The first works on the development of methods for corrosion protection of copper were carried out about 200 years ago by Sir Humphry Davy [17], who developed the principles of electrochemical protection of copper sheets of the hulls of ships [17]. Since then, various methods have been proposed to protect copper from corrosion; for example, sometimes to preserve a high-quality finish, protective metallic coatings of one or more of the following metals are used: tin, lead, nickel, silver, chromium, rhodium, or gold [9]. In some conditions, corrosion inhibitors are recommended as a protective measure. Nitrogen-containing compounds, such as those with an azole group in the molecule, such as benzotriazole, have proven to be the most effective for protecting copper. Sulfur-containing organic corrosion inhibitors, such as sodium diethyldithiocarbamate, are also effective [9]. But the most applicable method of protection is the use of organic coatings: for example, in some aggressive soils which contain bitumen or are polymeric [9], and in less severe conditions, paint and varnish [5,9] is used, or for copper and brass, transparent varnish is used [5]. In addition, the literature provides information on the protection of copper from corrosion by organosilicon coatings obtained by treating the copper surface with organosilanes [18,19,20,21,22,23,24], which often were used in a mixture with nitrogen- [22,23], sulfur- [18,21,24] and phosphorus- [22] containing compounds (inhibitors) or with the addition (filled) of graphene oxide nanoparticles [20]. However, despite many years of research efforts aimed at developing a means of anti-corrosion protection for copper, there is no reliable way to prevent copper corrosion to date. Thus, it is obvious that the development of new ways to increase the resistance of copper to the action of corrosive components of the environment is an urgent scientific and technical challenge.

Alkoxysilanes with the general formula RnSi(OC_2_H_5_)_4−n_ are environmentally friendly compounds [25,26] capable of adsorption on the surface of inorganic substrates (e.g., metals), hydrolysis with atmospheric water or adsorbing on the surface (according to reaction (1)), and polymerization to produce self-organizing siloxane nanolayers (Equation (1), Figure 1) [27,28].
(1)$$R_{n}Si(OC_{2}H_{5}{)}_{4-n}+(4-n)H_{2}O\to R_{n}Si{\left(OH\right)}_{4-n}$$

Such layers comprise reactive groups (R) capable of interacting with the reactive groups of a wide range of polymer coatings. This fact was successfully used to increase the strength and stability of “inorganic substrate-polymer” adhesive compounds, for example, in composite materials [28,29]. However, in addition to increasing adhesive interactions at the interface, siloxane layers can change the physicochemical properties of the metal surface, for example, by making it hydrophobic [28], and also prevent corrosion damage to metals [28,30].

This article is devoted to the study of the influence of organosilicon nanolayers obtained by treating the metal surface with organosilane solutions (without additives) on the electrochemical and corrosion behavior of copper.

The purpose of this work is to study the role of self-organizing surface nanolayers obtained from organosilanes in the inhibition of the anodic dissolution and corrosion of copper.

## 2. Materials and Methods

Copper of M006 grade (99.99%) meeting the requirements of the international standard [31] was used. The copper surface was modified with solutions of the following compounds: (a) organosilanes with general formula R_n_Si(OR’)_4−n_ (the manufacturer LLC “Silan”, Moscow, Russia):
-vinyltrimethoxysilane (VS) CH_2_=CHSi(OCH_3_)_3_;-γ-aminopropyltriethoxysilane (AS) NH_2_–(CH_2_)_3_Si(OC_2_H_5_)_3_;

(b) an organic corrosion inhibitor of the azole class, namely, 1,2,3-benzotriazole (BTA) C_6_H_5_N_3_ (its structural formula is shown in Figure 2) (The manufacturer LLC Enterprise "ROD", Moscow, Russia)**.**

The copper surface was modified both by solutions of the individual compounds (modifiers) listed above and their mixtures. The solvents used were water for organosilanes and their mixtures and ethyl alcohol for BTA and mixtures of BTA with organosilanes. The concentrations of the modifying components in solutions were 0.1–3.0% for organosilanes and 10 mM for BTA. The compositions of the modifying solutions are given in Table 1.

Rectangular copper specimens, dimensions 50 × 100 mm (4 × 6 in.) [32], 1.5 mm thick were used (Figure 3).

Since copper manufacturers often supply copper as wire rolls (Figure 4), in addition to rectangular specimens, specimens of copper rod wire with a diameter of 8 mm and a length of 10 cm were used.

Electrochemical studies were carried out on square specimens 10 × 10 mm in size, 1.5 mm thick (Figure 5). A current lead was soldered to the non-working surface of the specimen; the point where the current lead contacted the specimen was reliably insulated.

The surfaces of the specimens were treated with sandpaper and polished with diamond pastes of ASM 2/1 POM, ASM 1/0 POM, and ASM 0.5/0 POM grades [33,34]. After the treatment, the surface was degreased with ethyl alcohol and then chemically etched with 5% H_2_SO_4_ solution for 2 min. After that, the specimen was washed with water for 5 min, air dried, and modified with a modifying solution (Table 1) for 10 min. After modification, the specimen was placed in a solvent for 1 min to remove the excess of unreacted modifier. In addition to massive specimens, a quartz resonator with a thermally sputtered copper layer 1 µm thick underwent modification.

After modification, the specimens were weighed on AF-R220CE analytical scales (Shinko Denshi Co., Ltd., Tokyo, Japan) and examined by optical microscopy using an MMR-34 metallographic inverted microscope (“LOMO-MA”, St. Petersburg, Russia).

The amount of organosilane on the surface was determined gravimetrically by pesoquartz nano-weighting [35]. Massive copper specimens and a quartz resonator with a thermally sputtered copper layer (1 μm thick) were modified in parallel. Gravimetric measurements were performed on a thin layer of copper sputter deposited from a vacuum (10^−6^ mmHg) on the working element of the quartz resonator. The variation in the frequency of the quartz resonator was measured and the mass change upon deposition (adsorption) of organosilanes was calculated as follows (Equation (2)):(2)$$-\Delta m=\frac{\rho NS\times \Delta f}{f_{0}^{2}}$$
where *f_o_* is the main base frequency of 10,000 kHz; Δ*m* is the change in the silane mass (g); Δ*f* is the change in the frequency of the piezoquartz resonator (kHz); *N* is the frequency constant (for AT-cut crystals, *N* = 1670 kHz mm); *ρ* is quartz density, 2.65 g/cm^3^; *S* is the quartz working area equal to 0.72 cm^2^. 

The layer thickness was calculated from the adsorbate mass, assuming a uniform distribution of organosilane molecules on the surface.

The thickness of the organosilane layer was also estimated from gravimetric data by weighing the specimen on analytical scales before and after the modification. The layer thickness *h* (in µm) (Figure 6) was determined using Formulas (3)–(5) as follows:(3)$$V=Sh=\frac{\Delta m}{\rho }$$
(4)$$h(cm)=\frac{\Delta m}{\rho S}$$
(5)$$h\left(\mu m\right)=\frac{10000\times \Delta m}{\rho S}$$
where *V* is the volume of the surface layer (cm^3^), Δ*m* is the increase in the specimen mass after modification in an aqueous solution of an organosilane (g), *S* is the specimen area (cm^2^) (Figure 3), and ρ is the specific density of the organosilane (g/cm^3^).

In some cases, the thickness of the organosilane layer was converted to the number of molecular layers.

The accelerated corrosion tests of copper specimens were carried out in 0.1 M aqueous sodium chloride solution. During the tests in the solution, in some cases, copper corrosion was studied by scanning reflectometry [36,37] using a glass cylindrical cell with a smooth, even, optically transparent bottom. A specimen was placed on the bottom of the cell with the working surface downward, providing a gap between the metal surface and the bottom of the cell by means of a glass stopper. The angle between the metal surface and the scanner’s object glass was chosen in such a way that the image of the specimen in mirror-reflected light could be obtained. A standard Epson Perfection 3200 Photo scanner (Seiko Epson Corp. Suwa, Nagano, Japan) with an optical resolution of 3200 dpi was used. Metal corrosion was evaluated by the change in the area of the corroded surface during the test. In parallel with scanning reflectometry, the variation in the corrosion potential of the test specimens was continuously recorded for 10 days. The potential of the copper specimen was measured vs. a silver chloride reference electrode using an ARRA 109N digital multimeter (MGL APPA Corporation, Teipei City, Taiwan). The values of potential were converted to the standard hydrogen electrode (SHE) scale. Digital image processing commonly used for detailed studies of surface morphology [38] was employed to visualize the localized changes in the specimen’s reflectivity. Calculation of the degree of surface coverage with defects and the average size of an individual defect was carried out using an original digital image processing software written in Ruby 1.9.0 language and the RMagick 2.12.0 program (ImageMagick 6.5.6-8). Three-dimensional visualization of surface defects was performed using original software developed on the basis of the Surfer 9.0 program.

The effect of surface layers on the atmospheric corrosion of copper was studied by outdoor and accelerated corrosion tests. The outdoor corrosion tests were carried out at the test sites of the Moscow Corrosion Station (MCS) and the Northern Corrosion Station (NCS) of Frumkin Institute of Physical Chemistry and Electrochemistry of the RAS in accordance with the techniques reported elsewhere [32,39]. The Moscow corrosion station is located in the southwest of Moscow in an industrial-urban environment and is equipped in accordance with the requirements for test stations [32,40]. According to Ref. [41], the atmospheric corrosion category at the MCS with respect to zinc and copper is C3 (medium). The NCS is located on the coast of the Arctic Ocean (Barents Sea) in a moderately cold climate (69°07′ N, 36°04′ E) on a narrow (0.5–3.0 km wide) peninsula surrounded by sea water on three sides. The terrain is hilly. The Barents Sea does not freeze. The shortest distance from the sea to the test site is 300 m, and the site elevation above sea level is 32 m. The atmospheric corrosion category [41] at the NCS with respect to zinc and steel is C3 (medium), and with respect to copper is C4 (high).

In the southwest of Moscow, the specimens were mounted on test stands according to the recommendations given in Refs. [32,40]. Five specimens of each type of metal and all systems under study (both unmodified and modified) were installed. In addition, tests were carried out by accelerated corrosion methods at elevated temperature and humidity [42] and salt content in accordance with the requirements of the international standards [43,44]. The tests at elevated temperature and humidity were carried out in an MHK-408CL climate chamber (Terchy Environmental Technology Ltd., Taipei, Taiwan) with a working volume of 408 L. The temperature (t) was 60 °C, RH 95%. The temperature was maintained to within 0.2 °C, and humidity to within 0.2%. The tests with increased salt content were carried out in a KST-500 salt fog chamber (VacTime Ltd., Moscow, Russia), the working volume of which was 500 L, with a salt fog precipitation rate of 1.0–2.5 mL/h, a working temperature range of +(15–25) °C, and a humidity of 95%.

In some cases, accelerated corrosion testing methods used in the automotive industry were used in the tests [45,46,47], namely, the Volvo Indoor Corrosion Test or Volvo Cycle (Volvo-VICT) [46] and Hoogovens Cyclic Test (HCV) [47].

The tests were carried out according to the following scheme:

Step 1. 7 h RH 90%, t = 35 °C; 

Step 2. 1.5 h continuous linear RH variation from 90% to 45%, t = 35 °C; 

Step 3. 2 h RH 45%, t = 35 °C;

Step 4. 1.5 h continuous linear RH variation from 45% to 90%, t = 35 °C. 

Once a week Step 1 was replaced by:

Step 5. Specimens were removed from the chamber and immersed or sprayed with 1% salt solution for 1 h.

Step 6: Specimens were removed from the solution and excess solution was allowed to drain from the specimen for 5 min. After that, the specimens were returned to the chamber with 90% humidity and left for 7 h to dry (Steps 2, 3).

The composition of the salt solution was 0.5% NaCl + 0.1% CaCl_2_ + 0.075% NaHCO_3_.

The total duration of VICT and/or HCV tests was 12 weeks.

For testing copper rod specimens (Figure 4), test modes that simulate the conditions of metallic copper transportation were chosen. Briefly:

Mode 1, simulating the conditions of copper rod transportation by sea in the humid tropical climate zone: air temperature +40 ± 1 °C, relative humidity 98%, deposition rate of chloride aerosol 300 mg/m^2^ per day.

Mode 2, simulating the conditions of copper rod transportation and storage in a cold and temperate climate zone with cyclic temperature variation: the upper and lower values of the air temperature were +20 ± 3 °C and −15 ± 1 °C.

Mode 3. Comparative tests of corrosion protection means against the action of chlorides: air temperature +40 ± 1 °C; relative humidity 98%, immersion of specimens into 1% NaCl solution at 40 ± 1 °C for 1 h (Volvo VICT method).

Accelerated corrosion tests were performed on copper rod specimens 10 cm long (Figure 4). Specimens with different surface treatment were tested:untreated specimens (surface as received);specimens treated with sandpaper, 0 grade;specimens with “Libro” protective coating (Libro, Poland) that was applied by immersion in 5% (vol.) aqueous solution of concentrate at 65 °C followed by air drying at room temperature;specimens with GKZh hydrophobizing coating (JSC “GNIICHTEOS”, Moscow, Russia), applied by immersion in 10 vol.% solution of gasoline followed by air drying at room temperature;specimens with vinyl and amine-containing organosilicon coating (VS, AS and [VS + AS] mixture) applied by immersion into 1 vol.% aqueous solutions of organosilanes followed by air drying at room temperature.

Some specimens were tested in hermetically sealed 5 × 3 cm polyethylene bags, film thickness 0.1 mm (“bag” label), or in similar bags with two through holes (5 mm diameter; “perforated bag” label). Specimens were weighed on an AF-R22°CE analytical scale (Shinko Denshi CO., Ltd., Tokyo, Japan).

Tests were carried out in climate chambers Feutron KPK-630 (Feutron Klimatsimulation GmbH, Langenwetzendorf, Germany) (Modes 1 and 3) and MHK-408CL (Terchy Environmental Technology Ltd., Taipei, Taiwan) (Mode 2) for 31 days (744 h).

The test modes are presented in Table 2. 

During the tests, the specimens were placed horizontally in the chamber without touching each other.

Chloride precipitation (mode 1) was carried out by spraying 10% NaCl solution with a nozzle in a 0.4 m^3^ chamber. The rate of chloride precipitation was monitored on five flat 10 × 15 cm glass pieces placed next to the specimens.

Chloride precipitation by immersion (mode 3) was carried out by immersing the specimens in 1% NaCl solution once every 3 days for 1 h. The container with specimens was placed in a climate chamber at an air temperature of +40 ± 1 °C.

The specimens were inspected every 48 h. The changes on the surfaces of the specimens due to corrosion were noted and the specimens were photographed. The results were recorded in the test log.

Copper corrosion rate was measured gravimetrically from the difference in the mass of specimens before and after the tests. The specimens were weighed before the tests and after removal of corrosion products using AF-R220CE electronic analytical scales (SHINKO DENSHI Co., LTD., Tokyo, Japan). At the end of the tests (before weighing), corrosion products were removed from the surface of the specimens using the standard methods [39,45].

The mass loss of specimens per unit area [Δm], g/cm^2^, was calculated according to Equation (6) [46]:(6)$$\Delta m=\frac{m_{0}-m_{1}}{S}$$
where *m*_0_ is the mass of the specimen before testing in g;

*m*_1_ is the mass of the specimen after testing and removal of corrosion products in g;

*S* is the surface area of the specimen in cm^2^.

If necessary, the depth of corrosion penetration (change in thickness of a flat specimen Δ*L*) was calculated from the mass loss taking the specimen’s geometry into account:(7)$$\Delta L=\frac{\Delta m}{\rho }$$
where Δ*m* is the mass loss per unit area in g/cm^2^;

ρ is the metal density in g/cm^3^.

The corrosion depth (*K*, mm/year) was calculated from the depth of corrosion penetration:(8)$$K=\frac{365\times \Delta L}{10\times t}$$
where *t* is the test time (days). 

The kinetics of the corrosion process were determined in accordance with the criteria [47] from the degree of corrosion damage of the surface (*G*, %) using Equation (9) [46]:(9)$$G=\frac{\sum_{i-1}^{n}S_{i}}{S}*100$$
where *S_i_* is the area of a corrosion spot in m^2^;

*n* is the number of spots;

*S* is the surface area of the specimen in m^2^.

In addition, the degree of corrosion damage of the specimens was estimated visually, using digital image processing and evaluating the degree of corrosion in accordance with the requirements of the international standard [47].

Electrochemical measurements were carried out in a standard three-electrode cell using an IPC-Pro MF potentiostat (LLC “Volta”, St. Petersburg, Russia). The specimens were treated with grade “0” sandpaper and then additionally washed for 25 min in a “Sapphire—0.8 TC” ultrasonic bath (LLC “Sapphire”, Moscow, Russia) in a C_2_H_5_OH:C_7_H_8_OH = 1:1 mixture. To exclude the edge effects, the specimen after modification and air drying for 120 min was covered with a chemically resistant varnish, leaving an “open window” in such a way that the area of the electrode working surface was 1 cm^2^. The measurements were carried out in borate buffer (0.4 M H_3_BO_3_ + 5.5 mM Na_2_B_4_O_7_) with a pH of 6.7, with the addition of 0.1 M NaCl. A platinum electrode with an area of 1.2 cm^2^ was used as the auxiliary electrode. The potentials were measured relative to a silver chloride reference electrode; the measured potential values were converted to the standard hydrogen electrode (s. h. e.) scale. When using the electrochemical method of polarization curves and obtaining polarization curves of copper, the sweep rate of potential rate was 1 mV/s.

The critical potential of pit formation *E_pt_* (or the potential of local steel depassivation), i.e., the potential above which pitting dissolution of metal occurs and stable pits are formed, was determined from anodic polarization curves as a break in the curve at a point where a sharp increase in current was observed [48,49]. The inhibition activity of an organosilane was characterized by the value of *E_pt_* shift (Equation (10)) towards positive values [50] as follows:(10)$$\Delta E_{pt}=E_{pt-\mathrm{mod}}-E_{pt-f}$$
where *E_pt-mod_* and *E_pt-f_* are the pitting potentials with and without modification, respectively.

The inhibiting ability of the surface layer was evaluated using the corrosion inhibition coefficient γ [50] calculated by Formula (11) as follows:(11)$$\gamma =\frac{K_{0}}{K_{\mathrm{mod}}}$$
where *K*_0_ and *K_mod_* are the corrosion rates of the specimen before and after modification. 

Scanning electron microscopy (SEM) and X-ray spectral microanalysis (XRS) [51] were performed on a Camebax SX50 microanalyser (CAMECA SAS, Gennevilliers Cedex, France) using a Si(Li) solid-state detector. The surface morphology was studied by atomic force microscopy using a “Solver-Pro” atomic force microscope (“NT-MTD”, Zelenograd, Russia) in “ex situ” (in air) contact mode.

## 3. Results and Discussion

The processes of formation of surface organosilicon layers on copper surfaces were studied. Figure 7 shows microphotographs of initial flat copper specimens. As can be seen from the figure, deep cleaning scratches were observed on the surface after polishing (Figure 7a). Subsequent chemical etching resulted in leveling of the specimen’s surface and reduction in the surface roughness. 

Photographs of the specimens modified with 1% aqueous solution of VS are shown in Figure 8.

As can be seen from Figure 8, the so-called “tempering colors” appeared on the surface due to the interference of reflected light in the surface layer, which indicates the presence of a surface film. The “tempering colors” were more pronounced on the pre-polished specimen (Figure 8a). It can be seen that the color varied over the surface, i.e., the colors are visible on a few segments with areas from 0.002 to 0.0075 mm^2^. Almost no colors were seen on most of the surface. These data indicate an uneven distribution of the film after modification of the pre-grinded surface. If the surface was not only polished but also treated chemically (etching) before modification, the non-uniformity of color distribution was not observed (Figure 8b). At first glance, “tempering colors” were not noticeable at all. However, “tempering colors” could be detected on closer examination of the surface. However, they were evenly distributed, which may indicate that the vinylsilane film was distributed more uniformly over the surface.

Apparently, chemical etching results in surface smoothing by removing irregularities and defects unseen to the eye (even under a microscope) that provoke the uneven growth of the silane layer, which is observed on pre-polished but not etched surfaces. 

Preliminary (rough) estimations of the thicknesses of the surface layers from optical data using the Newton color shade method [52] showed that the thickness of the uniform film (Figure 8b) is small and does not exceed 50–100 nm. The “basic” vinylsilane film, which covers the entire surface of the specimen, has approximately the same (but somewhat smaller, not more than 50 nm) thickness. The thickness of irregular areas (“islets”) is much greater and reaches several hundreds of nanometers (up to 0.8–0.9 µm).

Estimation of silane films thickness by gravimetric measurements showed that after modification of the copper surface with vinylsilane, the thickness of the surface layer was about 168 nm (Table 3). This value reflects the average film thickness over the surface. Replacement of the vinyl-containing silane with the amine-containing one resulted in an almost 2-fold increase in thickness, up to 302 nm (Table 3). In this case, the formation of an irregular surface film was also observed. However, the amine-containing film is significantly thicker than the vinyl-containing one, probably because the surface reactions of condensation of silanol molecules (the result of organosilane hydrolysis in aqueous solution—reaction (12)) with metal surface groups (reaction (13)) and surface polymerization (reaction (14)) occur more completely, since the amino-containing compounds are catalysts of reactions (13) and (14), acting also as a self-catalyst [53].
(12)$$R_{n}Si{\left(OR^{\prime }\right)}_{4-n}+(4-n)H_{2}O↔R_{n}Si(OH{)}_{4-n}+(4-n)R^{\prime }OH$$
(13)$$Cu_{s}-OH+(OH{)}_{3}SiR\to Cu_{s}-O-Si{\left(OH\right)}_{2}R+H_{2}O$$
(14)$$n[Cu_{s}-O-Si{\left(OH\right)}_{2}R]\to [Cu_{s}-O-SiR(OH)-O-SiR(OH)-{]}_{n}+(n-1)H_{2}O$$

Modification of copper surface with a mixture of amino and vinyl silanes resulted in the formation of a uniform film on the surface (Figure 9).

The thickness of the layer determined from gravimetric data was about 900 nm (Table 3), which is almost three times greater than the thickness of the vinylsilane film and more than 5 times greater than that of the aminopropylsilane film. The reason for this is apparently a more complete surface polymerization of silane molecules during surface modification with the mixture and formation of not only linear but also cross-linked polymer fragments. Therefore, one may expect the formation of a denser film in the case of the AS + VS mixture for modification of the metal surface.

Atomic force microscopy (AFM) data confirm the conclusion that a non-uniform film is formed upon modification of copper surface by individual organosilanes (Figure 10a,b). In case of AS and VS (Figure 10b), one can observe thicker irregularities (“islets”) on the background of the uniform surface layer. Moreover, upon modification of the surface by VS, the film is obviously denser and thicker than in the case of AS.

Modification of the surface with a mixture of organosilanes or with a mixture of VS and BTA resulted in a uniform surface layer (Figure 10c,d) repeating the topography of the original surface. However, it is possible to distinguish thickened areas (“islets”) on the surface of copper modified with the mixture of vinylsilane and benzotriazole (Figure 10c) The diameter of such “islets” is approximately 0.1 μm, while the diameter of thickenings (“islets”) is several times larger (0.8–0.9 μm) in the case of surface modification with solutions of individual organosilanes.

Thus, modification of copper surfaces with individual organosilanes results in the formation of uneven surface films whose thickness does not exceed 300 nm. The use of the mixture of organosilanes, namely the mixture of vinyl and amine-containing silanes, provides the formation of relatively “thick” surface layers with a uniform thickness close to one micron (1000 nm).

The effect of organosilane surface layers on the electrochemical and corrosion behavior of copper in chloride-containing solutions was studied. Figure 11b shows the kinetics of the corrosion potential variation of massive copper specimens during the initial period of corrosion tests. In the beginning of the tests, the corrosion potential of unmodified copper shifts to the positive region (Figure 11a); however, after 100 min, the potential starts to decrease. After 1300 min of the experiment, the potential changes insignificantly (Figure 11). The presence of the organosilicon nanolayer on the surface caused a shift in the corrosion potential by 75 mV in the positive direction (Figure 11b), which may indicate its protective effect. In addition, the change in the corrosion potential of this specimen in the initial period of testing differed from that of pure (unmodified) copper. Namely, in the initial period (about 5 min), the corrosion potential decreased when the specimen was immersed into the corrosive medium (Figure 11b), then a slight shift in the potential to the positive values was observed. During a longer testing period, the potential of the specimen with an applied nanolayer changed similarly to that of pure copper (Figure 11a).

The variation in potential provides information on the processes occurring on the metal surface during the initial period of corrosion. In particular, is believed that a Cu_2_O layer grows on the surface of copper during corrosion in an aerated chloride-containing solution [54]. Moreover, copper dissolves as a result of corrosion, and the following reactions ((15) and (16)) occur on the metal surface [10]:
Anodic process:



(15)
$$Cu\to Cu^{+}+\bar{e}$$




Cathodic process:

(16)
$$1/2O_{2}+2H^{+}+2\bar{e}\to H_{2}O$$



The copper cations in the solution can react with chloride ions to produce copper chloride. According to [55], during corrosion, the potential of a metal covered with an oxide layer shifts to the cathodic region if the oxide is an n-type semiconductor and to the anodic region if it is a p-type semiconductor. Cu_2_O is believed to be a p-type semiconductor [56], while CuCl is an n-type semiconductor [57]. Thus, in the initial period, the potential shift to the cathodic region observed on pure copper indicates the formation of a CuCl layer on the surface. The increase in the corrosion potential (Figure 11) indicates the formation of an oxide, or, more likely, a mixed layer of corrosion products, Cu_2_O + CuCl, possibly with a varying ratio of components. The presence of an organosilicon layer on the surface results in a potential shift to the negative region in the initial period, indicating the growth of the Cu_2_O film, and only after some time an increase in the potential is observed that corresponds to the formation of a mixed film. Apparently, the organosilicon layer can limit the adsorption of chloride ions to the metal surface (which is confirmed by data in the literature on the ion-exchange character of interactions during local dissolution of metals in chloride-containing electrolytes [58]) and contribute to the inhibition of copper corrosion in chloride-containing media.

The electrochemical behavior of copper was studied using anodic polarization curves. Figure 12 present the anodic potentiodynamic polarization curves of copper (Figure 12, curve 1) and copper modified with organosilane-based compositions (Figure 12, curves 2–5). The shape of the polarization curves is typical of passivating metals [10,49].

To estimate the ability of the metal to undergo local dissolution (pitting) in accordance with Equation (10), the pitting potential was determined from the anodic curve break [48,49]. Table 4 shows the *E_pt_* values determined from the polarization curves given in Figure 12.

It can be seen from Table 4 that the surface organosilicon layers can inhibit the local anodic dissolution of copper. The organosilicon film formed by the mixture of vinyl- and aminosilanes inhibits pitting (and hence, local pitting corrosion of copper) most efficiently.

Thus, the study of the electrochemical behavior of copper has shown that, as a result of formation of thin surface organosilicon layers, a decrease in the intensity of both uniform and local corrosion of copper can be expected. Modification of copper surfaces with the mixture of vinyl- and aminosilanes is most promising in terms of corrosion inhibition.

The corrosion behavior of copper modified with organosilane-based formulations has been studied. Scanning electron microscopy (SEM) combined with X-ray spectral microanalysis (XrSMA) was used to study the composition of corrosion products on the copper surface formed during corrosion tests [51]. Figure 13 shows the microphotograph of the original surface of a copper specimen. The surface of metallic copper in air is covered with an air-oxide film (Figure 13a). The thickness of this film estimated from the intensities of X-ray-microanalysis spectra to be about 10 nm (Figure 13b). Exposure of pure copper specimens in a chloride-containing solution for 7 h results in the growth of a mixed film consisting of 73% Cu_2_O and 27% CuCl to a thickness of about 90 nm (the lighter areas in Figure 14). The surface is unevenly covered by a mixed oxide-chloride film occupying about 10% of the surface (the darker areas in Figure 15). The oxide part is an 11 nm thick Cu_2_O layer, which is close to the air-oxide film on the original copper specimen. After 21 h of testing, almost the entire surface is covered by a 180 nm thick mixed film with the composition 85% Cu_2_O—15% CuCl. Only 1% of the surface is covered with an air-oxide film. White islets with diameters of 0.8–2 µm were observed on the surface; the islets consist of a mixture of 80% Cu_2_O and 20% CuO.

Increasing the test time to 72 h results in the formation of a film of corrosion products on almost the entire surface of pure copper. This film is two-layered (Figure 15): there is a dense film with the composition 85% Cu_2_O + 15% CuO closer to the surface and a loose film consisting of a mixture of 55% Cu_2_O, 17% CuO, and 28% CuCl on the top. Further tests result in an increase in the thickness and ordering of the bilayer film (Figure 16). The near-surface layer covers the entire surface; it is about 250 nm thick and consists of a mixture of 87% Cu_2_O and 13% CuO. A porous layer is above the near-surface layer. It is close in composition to the lower one and has a thickness of 200–250 nm.

Thus, our data show that during the corrosion of pure copper in a weakly acid chloride-containing solution, a mixed film of Cu_2_O and CuCl first grows on the surface. However, the shift in the potential to the anodic region indicates that during the first few hours (2.5–3 h), a CuCl layer is formed on the surface [57]. After that, copper oxide Cu_2_O begins to grow and a mixed oxide-chloride layer with variable composition is formed on the surface of the corroding metal. In fact, after 7 h of exposure, the copper chloride content is higher than after 21 h. However, longer tests result in an increase in the fraction of copper chloride in the mixed layer.

The results of the “in situ” scanner-reflectrometric study of copper corrosion in 0.1 M sodium chloride solution are shown in Figure 17. Indeed, it can be concluded that in the first 3.5–5 h, a film of one type grows predominantly (Figure 17a), probably CuCl (film 1), but even after 2 h, a parallel growth of a film with other optical properties can be observed (Figure 17b–f). The entire surface is not covered by the film; a significant part of the surface is free from corrosion products. After 6 h, intense growth of another type of film, Cu_2_O (film 2), begins, and after prolonged tests (131 h), nearly the entire surface is covered with either film 1, film 2, or their mixture. It should be noted that after 21 h, the surface is 98% covered with the mixed oxide-chloride film (Figure 17g), while further tests result in 100% coverage of the surface with the film (Figure 17h) without an increase in the growth of the layer’s thickness. However, the film grows non-uniformly. Prolonged exposure of specimens in a chloride-containing solution results in the covering of the entire surface with a two-layer oxide-chloride film of corrosion products, with a dense layer located closer to the surface (Figure 17i) and a porous layer on top (Figure 17j).

As can be seen from Figure 17, cover layers (films with different compositions) are formed on the surface upon exposure to the chloride-containing solution: 18% of the surface is covered with the film after 2 h, 25% after 3.5 h, and after 5 h, the film covers 49% of the surface. After 6 and 7 h, the degree of surface coverage with the film becomes 56 and 83%, respectively. Two types of film can be identified on the surface: 1st, a film with a yellow color in reflected light, and 2nd, a film with a violet color. According to a rough estimate, the thickness of the film (both types) is small and does not exceed 80–100 nm. After 21 h of testing, almost the entire surface (98%) is covered by the film (Figure 17g), and after 72 h of testing, the film is not removed from the surface but continues to grow, occupying more than 99% of the surface area (Figure 17h). In longer tests (more than 21 h), a thicker film with non-uniform thickness is formed, with estimated thickness ranging from 0.8 to 2.4 µm. Prolonged exposure of the specimens to the chloride-containing solution results in the covering of the entire surface with a two-layer oxide-chloride film of corrosion products, with a dense layer located closer to the surface (Figure 17i) and a porous layer on the top (Figure 17j).

Thus, the presence of self-organizing organosilicon layers on the copper surface results in inhibition of copper corrosion. Figure 18 shows the kinetics of CuCl film formation at the initial stage of testing. It can be seen that the degree of inhibition depends on the thickness of the nanolayer. With a layer thinner than 2 molecular layers, a slight decrease in the corrosion rate was observed (Figure 18, Table 5). The increase in the layer thickness up to 3.8 molecular layers upon modification of the metal with 5 × 10^−2^ M VS resulted in the maximum inhibition effect: the corrosion rate decreases 4.2-fold. Further thickness growth does not result in an increase in the inhibition effect, and the corrosion rate even increases (Table 5). Apparently, an ordered organosilicon nanolayer 3.0–3.8 molecular layers thick is formed on the metal surface. An increase in the number of adsorbed molecules results in an increase in the layer thickness but degrades its regularity and protective properties. A similar effect was observed [59] concerning the adsorption of organosilanes from a gas flow on iron surfaces. Thus, the ordered layer closest to the copper surface has the strongest corrosion inhibition effect.

The experiments where the duration of corrosion tests was increased to 156 h showed that the vinylsiloxane nanolayer maintained its protective effect. Figure 19 demonstrates the effect of vinylsiloxane layer thickness on the corrosion behavior of copper in 0.1 M NaCl. It can be seen that the copper surface is almost completely (98%) covered by a 250 nm thick film of corrosion products consisting of a mixture of 86% Cu_2_O and 14% CuCl (Figure 20a region 2, Figure 20b). The presence of the organosilicon layer on the surface results in a decrease in metal corrosion. The layer with a thickness of 3.8 molecular layers is the most efficient (Figure 19); in this case, 45% of the surface is free from the two-layer oxide-chloride film of corrosion products (Figure 20a, region 1; Figure 20b; Figure 20a, region 2; Figure 20c, region 3; Figure 20d) and is covered only by a Cu_2_O film 37–40 nm thick. This indicates that the vinylsiloxane layer hinders the adsorption of chloride ions on copper surface. Unlike in the initial testing stage, the increase in the thickness of the vinylsiloxane layer (above 3.8 molecular layers) does not result in an increase in corrosion protection efficiency (Figure 19).

The kinetics of corrosion development (mass loss) on copper in sodium chloride solution are shown in Figure 21. Modification of the copper surface with individual silanes practically did not affect the corrosion behavior of the metal (Figure 21, curves 2, 3); the corrosion rate of copper covered with organosilane films was 13.42 and 13.53 μm/year after 90 days of tests, while unmodified copper corroded at a rate of 13.48 μm/year (Figure 22). Modification of the copper surface with a mixture of AS and VS resulted in the inhibition of copper corrosion—the corrosion rate decreased more than 1.5-fold, to 8.3 μm/year (Figure 22).

Unlike with individual organosilanes, where modification with a mixture of organosilanes gives a uniform densely packed “thick” layer on the surface, a thinner (Table 3) and non-uniform in thickness film was formed (Figure 10, Table 3). Thus, it can be expected that the thickness and density (degree of cross-linking) of the surface layer are responsible for the inhibition of copper corrosion.

In addition to uniform corrosion, local corrosion of metal was studied using scanning reflectometry. Figure 23 shows the kinetics of covering the copper surface with local corrosion defects, and Figure 24 is a 3D image of the surface coverage with corrosion defects.

As can be seen from Figure 23 and Figure 24, the surface organosilicon layer provides a decrease in the copper local corrosion intensity. The films formed after surface modification with the mixture of vinyl- and aminosilanes, where relatively thick densely cross-linked layers formed on the surface, were the most efficient.

Thus, the effect of organosilicon self-organizing layers on copper corrosion has been studied. It was found that the protective effect of the film depended on the thickness of the surface layer. The maximum efficiency of protection was observed after modification of the surface with individual silanes at a thickness of 3.8 molecular layers (2.7–3.0 nm) and after modification with the mixture of vinyl- and aminosilanes that gave films with a thickness of 900 nm–1 µm. In both cases, a dense layer formed on the surface hinders the adsorption of chloride ions. In the case of the mixture of silanes, the dense layer significantly reduces the rate of interaction of chloride ions with the surface copper atoms, reducing the formation of copper chloride compounds and, therefore, effectively protecting copper from corrosion in chloride-containing electrolytes.

The effect of organosilicon surface layers on the atmospheric corrosion of copper was studied. For this purpose, accelerated corrosion tests were carried out in climate chambers at elevated temperature and humidity and in a salt spray chamber at elevated sodium chloride content in the atmosphere. Moreover, outdoor corrosion tests were carried out in an urban industrial atmosphere.

The kinetics of local atmospheric corrosion development under the conditions corresponding to the requirements of the Volvo-VICT industry standard [45,46] are shown in Figure 25. Figure 26 demonstrates the appearance of specimens after 45 days of testing under the same conditions.

One can see from Figure 25 and Figure 26 that organosilicon surface layers efficiently inhibit the local corrosion of copper caused by the combined action of sodium chloride and elevated temperature and humidity. Apparently, when the metal is modified with vinylsilane or a mixture of VS and AS, dense cross-linked polymer films are formed on the surface, which, despite their relatively small thickness (6 and 5 molecular layers for VS and the [VS + AS] mixture, respectively), can efficiently inhibit the atmospheric local corrosion of copper.

Accelerated corrosion tests of copper rods (shown in Figure 4) were carried out under conditions simulating the conditions of copper transportation. Based on analysis of the literature on the impact of climatic and aerochemical factors on copper corrosion, the accelerated corrosion test modes described in Section 2 were selected (Table 2). 

As follows from an analysis of data in the literature [60,61,62,63] on the effect of climatic and aerochemical factors on atmospheric corrosion of copper, the one-year mass losses of copper in a humid tropical climate in the rural atmosphere range from 6 to 9 g/m^2^. The calculated copper corrosion rates in the climate chamber at an air temperature of 40 °C and a relative humidity of 98% (without chloride spraying) vary from 5.6 g/(m^2^·year) over 13 days of tests to 8.9 g/(m^2^·year) over 32 days. In the humid tropical marine atmosphere, the corrosion rate increases significantly, and the annual mass losses vary mainly in the range of 20–40 g/m^2^. The corrosion rate of protected copper rod specimens in tests in mode No.1 (see Section 2) amounts to 30–32 g/(m^2^·year). 

Thus, mode No.1 adequately simulates the conditions of copper rod corrosion in a humid tropical climate: without chloride spraying to mimic a clean atmosphere, with chloride spraying to mimic a marine atmosphere. Accordingly, the results of testing the corrosion protection means should match their efficiency under the outdoor conditions of a humid tropical marine atmosphere. However, it should be taken into account that the rate of chloride precipitation under natural conditions depends on the distance from the coastline, terrain features, wind speed and direction, and other factors [60,63], so it is impossible to give a quantitative prediction of copper corrosion rate and service life of protection means without accurate climatic and aerochemical characterization of the place of operation of the metal or metal item (structure).

The one-year mass losses of copper in cold and moderate climates in rural atmospheres range from 2.8 to 6.5 g/m^2^. The calculated corrosion rates of individual copper rod specimens in mode No. 2 tests in the climate chamber vary from 2 to 5 g/(m^2^·year) with a mean value of 3.4 g/(m^2^·year). Hence, mode No. 2 satisfactorily simulates the conditions of copper corrosion in a rural, tentatively clean atmosphere containing less than 20 µg/m^3^ of sulfur dioxide in the air in cold and temperate climates. 

In mode No. 3, intended for comparative testing of products for corrosion protection, the corrosion rate of protected specimens ranges from 71 to 82 g/(m^2^·year) with a mean corrosion rate of 76 g/(m^2^·year). Hence, the acceleration factor of mode No. 3 is 2.5 ± 1 relative to the most common values of copper corrosion rate in the humid tropical marine atmosphere. It should be taken into account that the duration of the tests should be no less than 30 days. In shorter tests, the corrosion rate of copper may decrease. 

To increase the acceleration factor, one can increase the frequency of immersing the specimens into the NaCl solution (up to once a day) and the concentration of this solution (up to 3%). However, it should be taken into account that the greater the acceleration factor of the tests, the less the results of accelerated tests match those of in situ tests for estimation of the efficiency of means for corrosion protection. The appearance of copper rod specimens after accelerated corrosion tests is shown in Figure 27, and the gravimetric results of accelerated tests are provided in Table 6 and Table 7. 

One can see from Figure 28 and Table 6 and Table 7 that the main role in the protection of copper from corrosion belongs to the isolation (sealing) of the metal from the environment. In fact, the corrosion rate of unmodified copper in sealed and perforated packages differs more than 4-fold (Figure 12). Treatment of the metal surface with a hydrophobizing formulation (GKZh) also led to a decrease in mass loss (Table 6 and Table 7). Modification of copper rods with solutions of individual organosilanes, like in the case of tests of flat specimens in solution, was less efficient than modification with a mixture of vinyl- and aminosilanes (Table 6 and Table 7, Figure 28). Copper modified with a mixture of organosilanes placed in a perforated bag corrodes more than three times more slowly than copper placed in a sealed bag, showing the highest inhibitory ability of this mixture among all the systems studied. Apparently, after modification with the silane mixture, a dense cross-linked polymer film is formed on the surface, which prevents the penetration of corrosion-activating agents to the metal surface and, hence, effectively inhibits the corrosion of copper, even in a perforated (non-hermetic) package. 

Accelerated corrosion tests were carried out on flat copper specimens in a climate chamber and a salt spray chamber. In the salt fog chamber, copper corrodes locally, since sodium chloride sprayed in the test chamber precipitates on the surface of the specimen, and after 1 day of testing, blue-green (turquoise) islets (“drops”), which apparently consist of a mixture of copper(II) hydroxide and chloride, are observed on the surface (Figure 29a). Apparently, an elevated content of chloride ions that are activators of local corrosion of the metal exists on the surface under these “drops”. Therefore, one can expect an intense development of corrosion defects under these drops, which was in fact observed after removal of the “drops” of the corrosive copper chloride/hydroxide solution—more than 60% of the surface is occupied by corrosion damage (defects) (Figure 30).

Preliminary modification of copper surfaces with organosilanes leads to the inhibition of the corrosion process. Figure 31 shows photos of copper specimens modified with organosilane-based formulations. From Figure 31, it can be seen that modification of the copper surface with solutions of individual silanes leads to inhibition of corrosion, as evidenced by the results of 8-day accelerated corrosion tests in a salt spray chamber: 40% and 27% of the surface is damaged by corrosion after preliminary modification with AS and VS solutions, respectively (Figure 31a,b). The surface film formed after modification with a mixture of vinyl- and aminosilane inhibited copper corrosion most efficiently. Single defects occupying about 12% of the specimen’s surface were observed (Figure 31c).

The data of gravimetric and optical-microscopic treatment of corrosion results confirm the conclusion about the corrosion-inhibiting effect of surface organosilicon layers. Figure 32 shows data on the corrosion rates of copper modified with organosilane-based formulations after 8-day tests in a salt spray chamber (RH 96%, t = 35 °C, 5 h of NaCl spraying). 

One can see from Figure 32 that, despite the fact that modification of copper surfaces with solutions of all the formulations studied results in inhibition of metal corrosion, solutions of individual organosilanes (VS and AS) affect the corrosion of copper insignificantly. A slight decrease in the corrosion rate by a factor of 1.1 and 1.2 for AS and VS, respectively, is observed. A stronger inhibition was provided by their mixtures. However, contrary to expectations, the mixture of a vinyl-containing silane with a known copper corrosion inhibitor [50], which effectively inhibits the corrosion of other metals [28,64], was not the most efficient, as it reduced the corrosion of copper only 1.4-fold and was inferior to the mixture of vinyl- and aminosilanes, which formed a dense cross-linked polymeric surface film and reduced the corrosion rate of copper by a factor of 1.7.

An optical microscopic assessment of the intensity of local corrosion was carried out. We estimated the degree of surface damage by corrosion defects and the parameters of these defects after the removal of the corrosion products. Figure 33 shows the appearance of a copper specimen after 8 days of accelerated corrosion tests in a salt spray chamber after removal of the corrosion products, while Figure 34 and Figure 35 list the parameters of the observed defects.

Accelerated corrosion tests in a climate chamber (at elevated temperature and humidity) caused uniform corrosion occurs on copper in the course of the tests, and the metal surface was covered with corrosion products that, judging by the color, apparently consisted mainly of copper(II) oxide (Figure 36). Preliminary modification of copper surfaces with solutions of organosilanes decreased the intensity of the corrosion processes (Figure 37). In fact, while in the case of unmodified copper, almost the entire surface is covered with a layer of corrosion products after 7 days of tests (Figure 37a), even when the surface is modified with solutions of individual silanes, the major fraction of the specimen surface is not covered with corrosion products: corrosion processes occur near the edge of the specimens (Figure 37b) and then apparently spread to the rest of the surface (Figure 37c). The degree of surface coverage with corrosion products after 7 days of testing was 7% and 12% in the case of VS and AS, respectively. Surface modification with a solution of a mixture of vinyl- and aminosilanes provided the maximum anticorrosion effect. After 7 days of accelerated tests, traces of corrosion were observed only on the edges of the specimens (Figure 37d), and corrosion products occupied no more than 2% of the specimen surface.

The results of gravimetric estimation of corrosion of copper specimens in a climate chamber are presented in Figure 38. 

It can be seen from Figure 38 that under elevated temperature and humidity conditions, as well as under chloride spraying conditions, the surface silane layers inhibit copper corrosion. The films formed after modifying the surface with a solution of a mixture of vinyl- and aminosilanes are the most efficient. In the case of this mixture, the corrosion rate over 32 days of testing decreased more than by a factor of 1.5 (more exactly, 1.6) (Figure 38). Thus, the results of accelerated tests showed that surface organosilicon films inhibit the atmospheric corrosion of copper in artificially created corrosive atmospheres.

To perform a more comprehensive study of the atmospheric corrosion of copper and the effect of surface cover layers on it, outdoor corrosion tests were carried out in urban industrial (Moscow Corrosion Test Station) and moderately cold marine (Northern Corrosion Test Station) atmospheres.

Figure 39 and Figure 40 show the results (corrosion rates after one-year full-scale tests), and Table 8 and Table 9 show the inhibition coefficients of atmospheric corrosion of copper. It can be seen from Figure 39 and Figure 40 and Table 8 and Table 9 that varying the composition of the modifying mixture (namely, changing the solvent from water to ethanol and varying the concentrations of some mixture components) does not play a fundamental role in the inhibition of atmospheric corrosion of copper. Moreover, it can be confidently stated that all the modifying formulations studied provide the formation of surface layers that decrease the rate of atmospheric corrosion of copper when exposed to the natural atmosphere (Figure 39 and Figure 40). The greatest efficiency, like in the case of accelerated tests, was demonstrated by the films formed after surface modification with an aqueous solution of a mixture of vinyl- and aminosilanes (Table 8 and Table 9), which reduced the corrosion rate more than 5- and 15-fold (compared to unmodified copper) in the urban and marine atmospheres, respectively.

Thus, the organosilicon surface layers efficiently inhibit the atmospheric corrosion of copper not only in artificial but also in natural atmospheres. 

## 4. Conclusions

It has been established that preliminary modification of the copper surface with aqueous solutions of individual organosilanes (both vinyl- and amino-containing) leads to the formation of nanosized polymer surface layers with thicknesses not exceeding 300 nm. Treatment of the copper surface with an aqueous solution of a mixture of vinyl- and amino-containing organosilanes leads to the formation of “thicker” (~1 micron thick) and uniformly (by thickness) distributed siloxane layers over the surface.Copper corrosion in a weakly acidic chloride-containing electrolyte was studied. The composition of the layer of corrosion products formed on the surface during testing was determined. It was shown that a copper chloride film was formed on the surface during the first 5–7 h of testing. An increase in the duration of holding the samples in the solution led to the growth of a two-layer film of corrosion products on the surface: the first layer was an ordered film consisting of 87% Cu20 and 13% CuCI and the second layer was a loose porous film of similar composition.The effect of vinylsiloxane self-assembled nanolayers on copper corrosion was studied and it was shown that the surface organosilicon layers formed on copper during modification effectively inhibit anodic dissolution and corrosion (including local) of copper in aggressive electrolytes, as well as in artificial and natural atmospheres. It was found that the protective effect of the nanolayer depends on its thickness. The maximum protection efficiency was observed at a thickness of 3.8 molecular layers, at which the densest layer is formed, hindering the adsorption of chloride ions and significantly reducing the rate of their interaction with surface copper atoms.

## Figures and Tables

**Figure 1 polymers-16-03066-f001:**
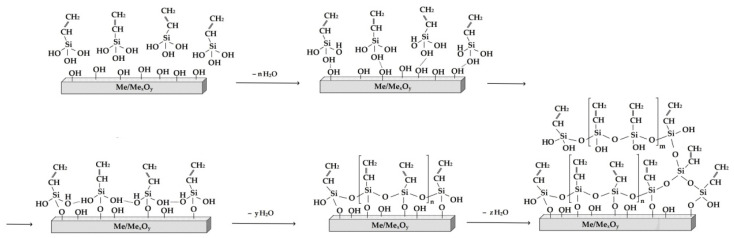
Scheme of formation of a surface organosilicon polymer layer on a metal.

**Figure 2 polymers-16-03066-f002:**
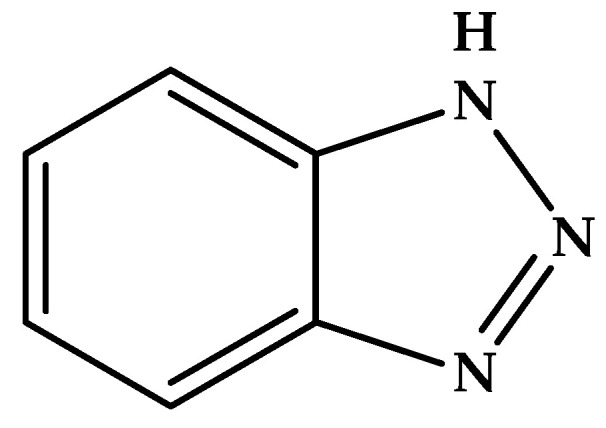
Structural formula of 1,2,3-benzotriazole.

**Figure 3 polymers-16-03066-f003:**
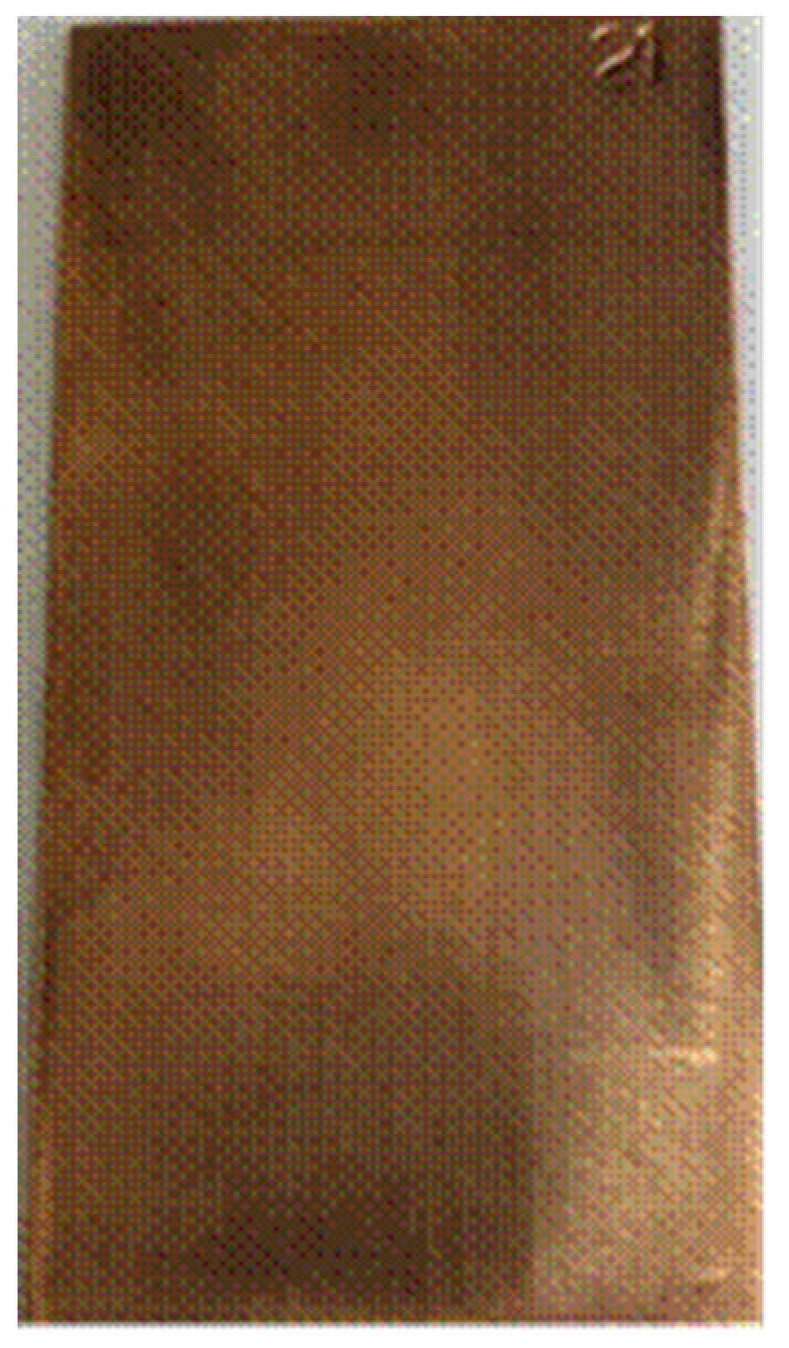
Appearance of a specimen for corrosion studies.

**Figure 4 polymers-16-03066-f004:**
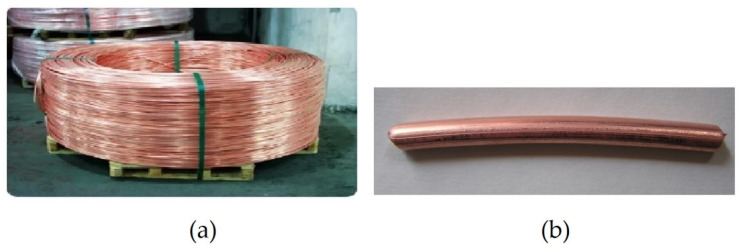
Copper as supplied by the manufacturer. (**a**) in the form of a rolled rod; (**b**) appearance of a specimen of copper wire for corrosion tests.

**Figure 5 polymers-16-03066-f005:**
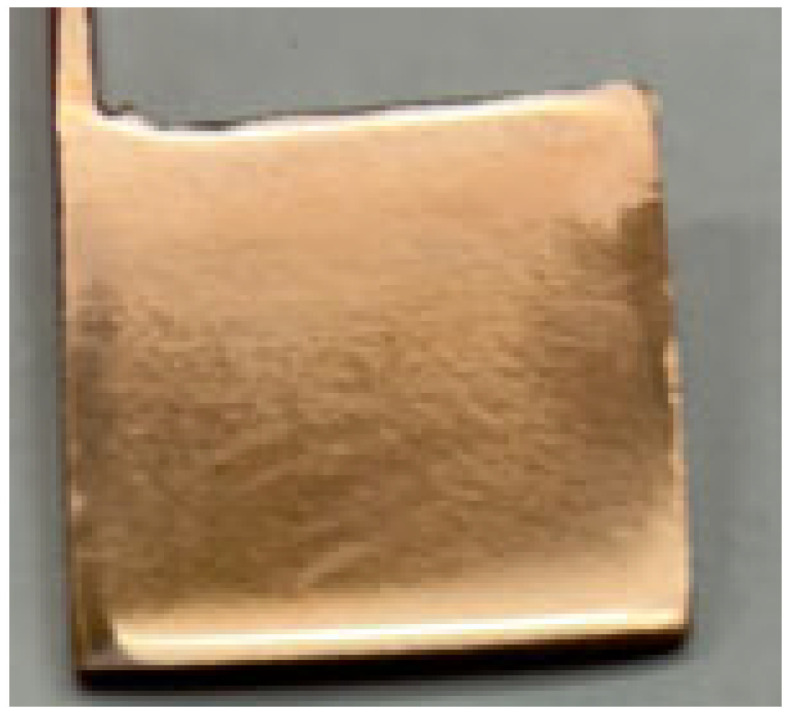
Appearance of the specimen for electrochemical measurements.

**Figure 6 polymers-16-03066-f006:**
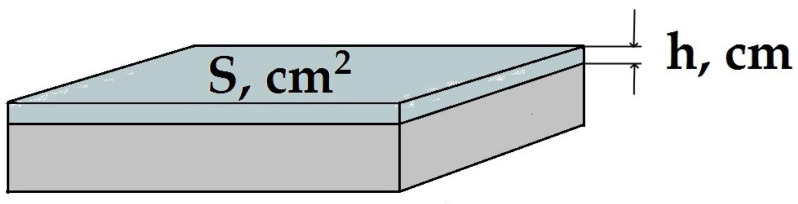
Parameters taken into account to determine the film thickness from gravimetric data.

**Figure 7 polymers-16-03066-f007:**
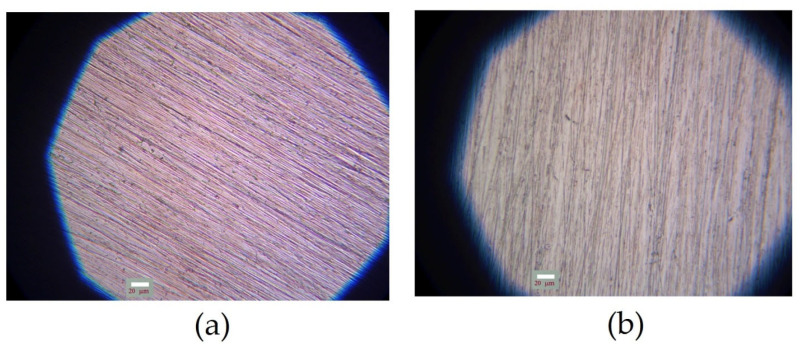
Microphotograph of the original copper specimen after surface pretreatment. (**a**) preliminary polishing; (**b**) preliminary polishing followed by chemical etching of the surface. Optical microscopy, magnification ×40.

**Figure 8 polymers-16-03066-f008:**
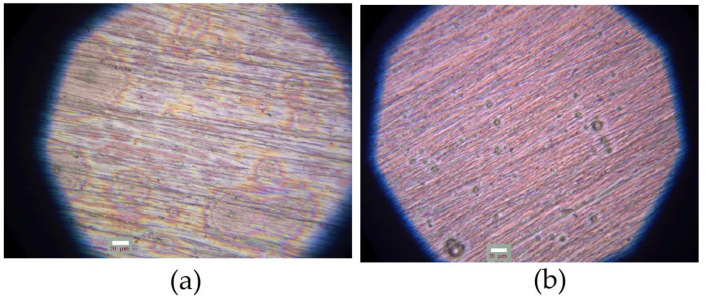
Microphotographs of copper specimens after surface modification with 1% aqueous solution of vinyltrimethoxysilane. Before modification, the surface was pre-treated: (**a**) preliminary polishing; (**b**) preliminary polishing followed by chemical etching of the surface. After modification, air drying at room temperature for 60 min was performed. Optical microscopy, magnification ×40.

**Figure 9 polymers-16-03066-f009:**
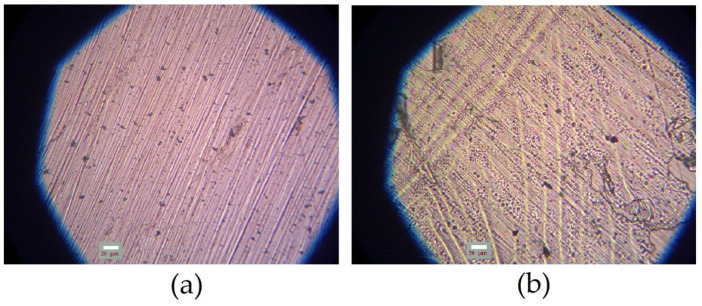
Microphotograph of copper specimens after modification with an aqueous solution of the [1% vinyltrimethoxysilane + 1% aminopropyltriethoxysilane] mixture. The sample exposure time in the solution: (**a**) 10 min; (**b**) 5 min. Optical microscopy, magnification ×40.

**Figure 10 polymers-16-03066-f010:**
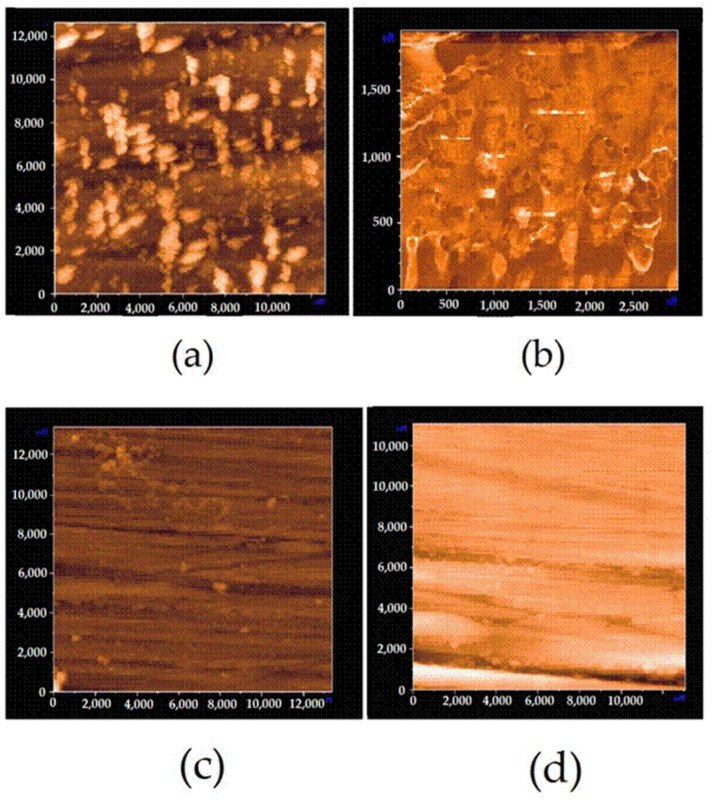
AFM images of the copper surface modified with (**a**)—an aqueous solution of AS; (**b**)—an aqueous solution of VS; (**c**)—an aqueous solution of the mixture [VS + 10 mM BTA]; (**d**)—an aqueous solution of the mixture [1%VS + 1% AS].

**Figure 11 polymers-16-03066-f011:**
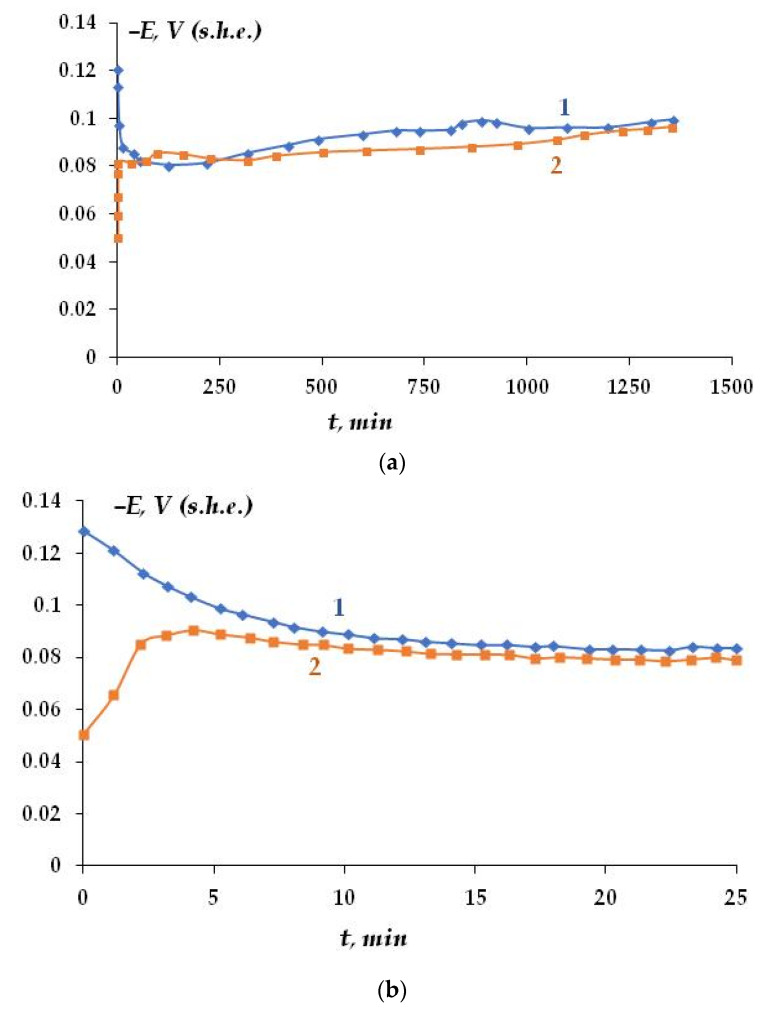
(**a**) Variation in the corrosion potential of copper (1) and copper with a surface vinylsiloxane nanolayer (168 nm thick) (2) during the initial period of corrosion tests in 0.1 M NaCl, pH 4.0.-Ttest period 1280 min.; (**b**) variation in the corrosion potential of copper (1) and copper with a surface vinylsiloxane nanolayer (168 nm thick) (2) during the initial period of corrosion tests in 0.1 M NaCl, pH 4.0; initial period of tests (the first 25 min).

**Figure 12 polymers-16-03066-f012:**
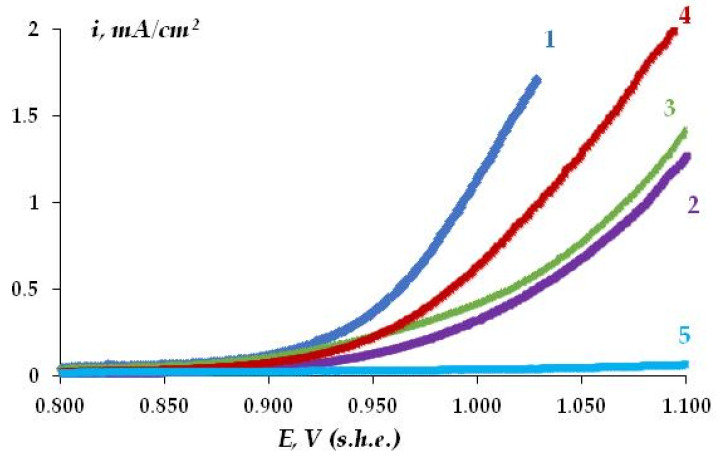
Anodic polarization potentiodynamic curves of unmodified copper (1), and copper modified by organosilanes (2–5): (2) 2% aqueous solution of VS; (3) 1% aqueous solution of AS; (4) [1% VS + 10 mM BTA] mixture; (5) [1% VS + 1% AS] mixture 0.1 M NaCl, pH 6.2. The potential scanning rate—0.1 mV/s.

**Figure 13 polymers-16-03066-f013:**
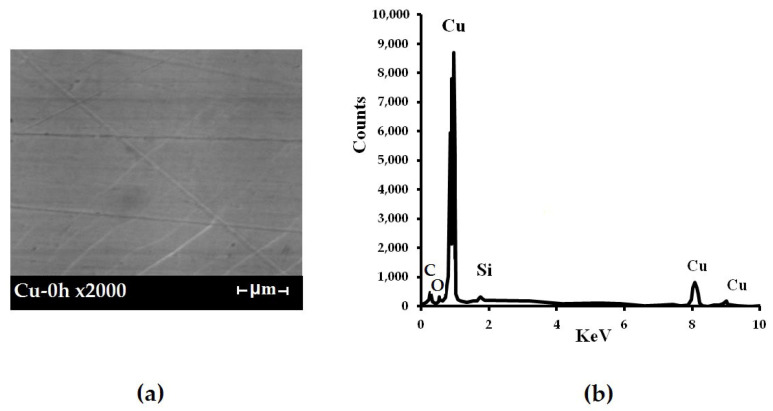
Copper surface before tests. (**a**) micro-photograph of the surface region, (**b**) XrSMA spectrum of the surface area modified with vinylsilane.

**Figure 14 polymers-16-03066-f014:**
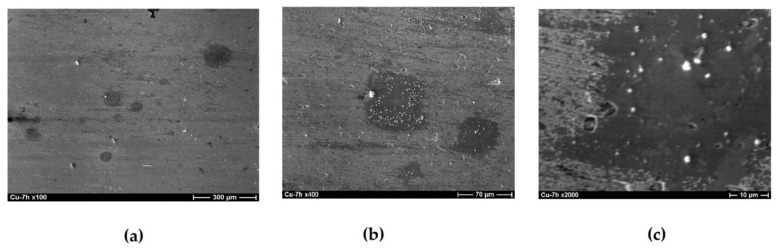
Copper surface after 7 h of corrosion tests in 0.1 M NaCl, pH 4.0. (**a**) magnification ×100; (**b**) magnification ×800; (**c**) magnification ×2000.

**Figure 15 polymers-16-03066-f015:**
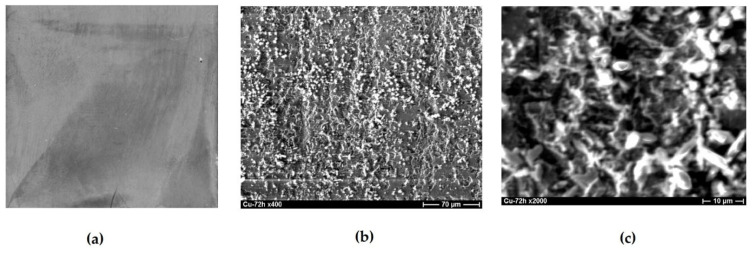
Copper surface after 72 h of corrosion tests in 0.1 M NaCl, pH 4.0. (**a**) full-size photograph of the specimen, (**b**) microphotograph with magnification ×400, (**c**) microphotograph with magnification ×2000.

**Figure 16 polymers-16-03066-f016:**
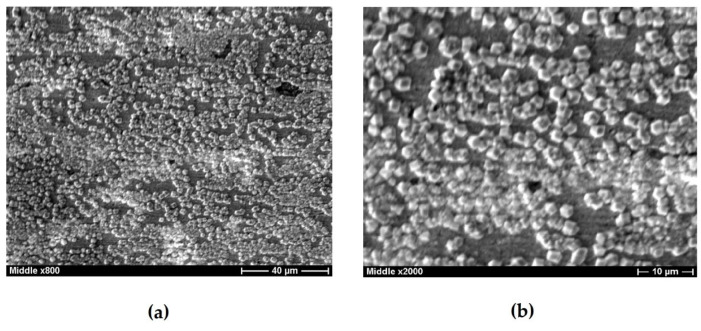
Copper surface after 156 h of corrosion tests in 0.1 M NaCl, pH 4.0. (**a**)—microphotograph with magnification ×800, (**b**)—microphotograph with magnification ×2000.

**Figure 17 polymers-16-03066-f017:**
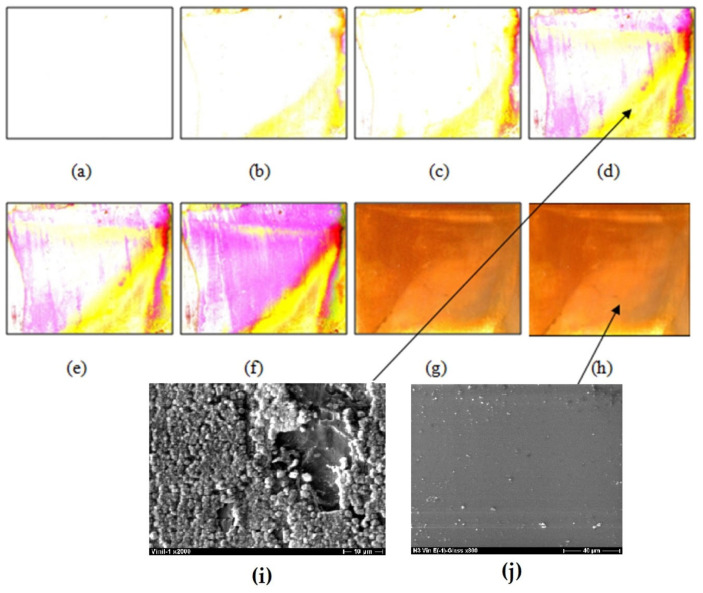
Change in pure copper surface during corrosion tests: (**a**) 0 h; (**b**) 2 h; (**c**) 3.5 h; (**d**) 5 h; (**e**) 6 h; (**f**) 7 h; (**g**) 21 h; (**h**) 72 h (0.1 M NaCl); (**i**) SEM microphotograph of the surface (shown in **d**) after 5 h of testing; (**i**,**j**) SEM microphotograph of the surface area (shown in **d**,**h**) after 5 and 72 h of testing, 0.1 M NaCl, pH 4.0. Arrows → show an enlarged (SEM format) image of copper surface areas.

**Figure 18 polymers-16-03066-f018:**
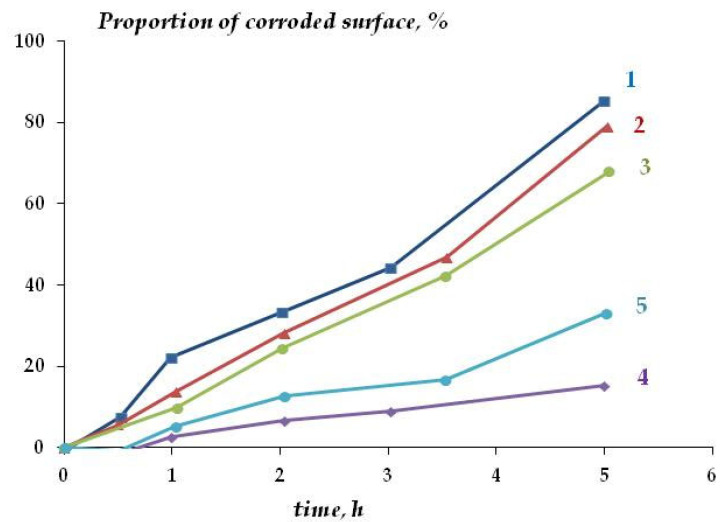
Effect of vinylsiloxane surface layers on CuCl film formation during the initial period of corrosion tests on pure copper (1) and copper coated with a vinylsiloxane nanolayer with a thickness of 0.86 (2), 1.52 (3), 3.81 (4), and 19.31 (5) molecular layers.

**Figure 19 polymers-16-03066-f019:**
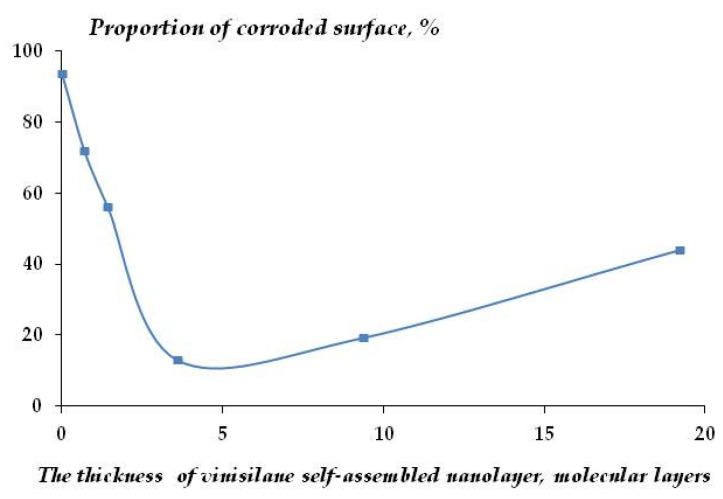
Effect of vinylsiloxane self-organizing layers on copper corrosion after 156 h of corrosion tests, 0.1 M NaCl, pH 4.0.

**Figure 20 polymers-16-03066-f020:**
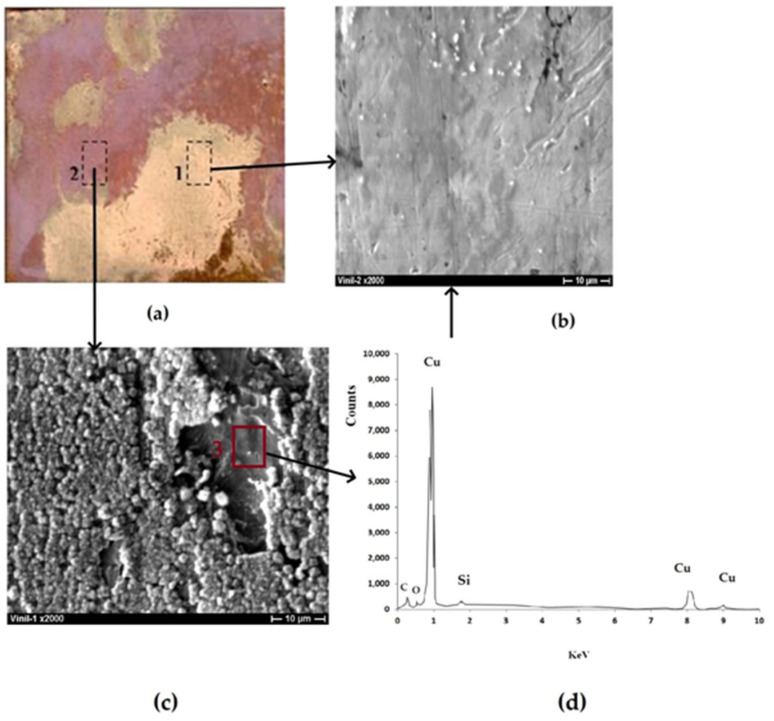
Copper surface with a vinylsiloxane film (3.8 molecular layers) after 156 h of corrosion testing in 0.1 M NaCl, pH 4.0. (**a**) full-size photograph of the specimen; (**b**) surface area 1 (**a**), magnification ×2000; (**c**) surface area 2 (magnification ×2000; **d**) XRS spectrum corresponding to region 1 (**a**) and region 3 (**c**). Image (**b**) is an enlarged section 
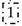
 of figure (**a**); image (**c**) is an enlarged section 
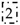
 of figure (**a**); spectrum (**d**) is the spectrum taken from section 
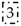
 of figure (**c**).

**Figure 21 polymers-16-03066-f021:**
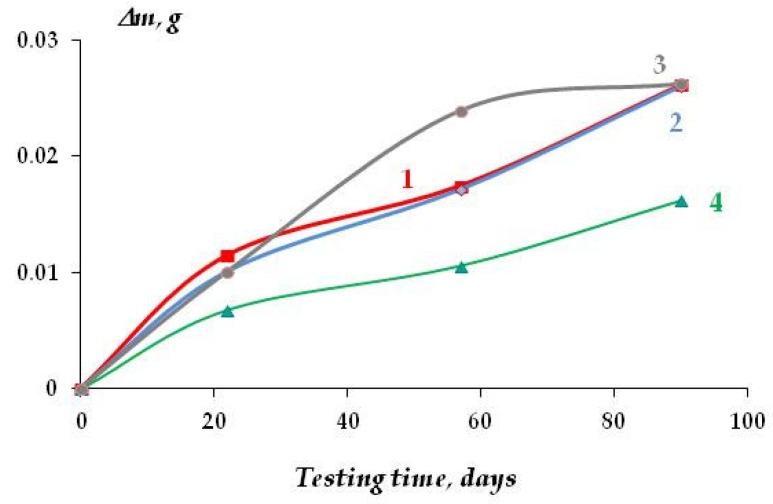
Kinetics of copper mass loss due to the metal corrosion in 0.1 M NaCl, pH 6.2. 1—unmodified copper; 2—copper modified with VS; 3—copper modified with AS; 4—copper modified with a solution of the mixture [VS + AS].

**Figure 22 polymers-16-03066-f022:**
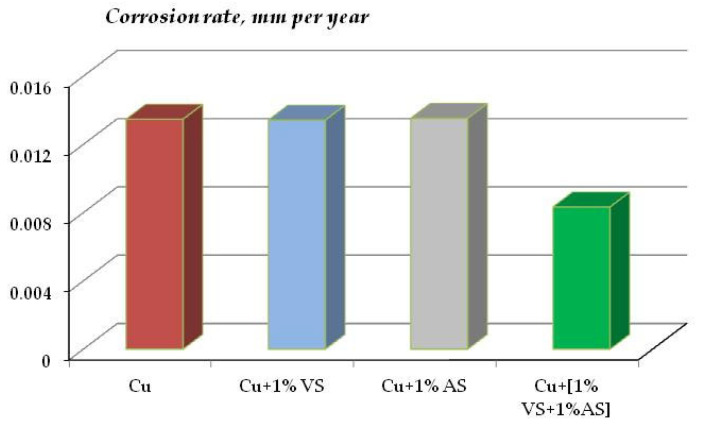
Corrosion rate of copper coated with silicon-organic self-organizing layers, 0.1 M NaCl, pH 6.2. The test duration was 90 days.

**Figure 23 polymers-16-03066-f023:**
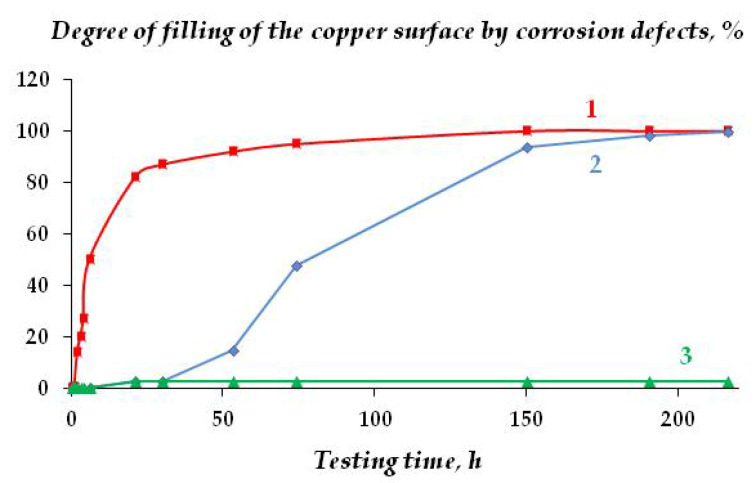
Kinetics of copper surface coverage with local corrosion defects during corrosion in chloride-containing solution: 1—unmodified metal; 2—copper modified with VS; 3—copper modified with the VS + AS mixture. “In situ” scanner reflectometry, 0.1 M NaCl. pH 6.2.

**Figure 24 polymers-16-03066-f024:**
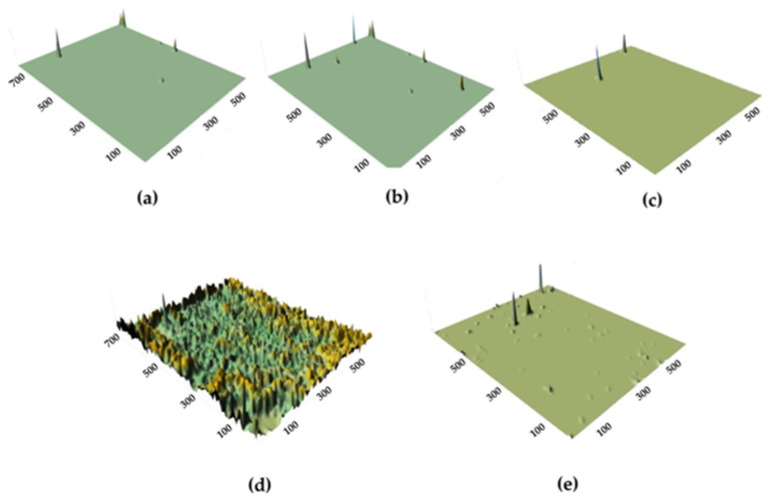
Three-dimensional representation of copper surface coverage with localized corrosion defects during corrosion. (**a**,**d**) unmodified metal; (**b**) copper modified with VS; (**c**,**e**) copper modified with the VS + AS mixture; 0.1 M NaCl, duration of tests: (**a**–**c**) 5 h; (**d**,**e**) 22.5 h. Processing of scanning reflectometry data.

**Figure 25 polymers-16-03066-f025:**
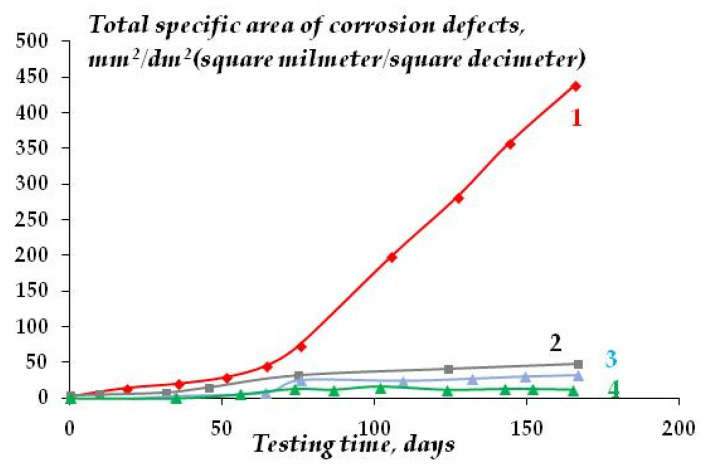
Kinetics of the development of corrosion defects on copper surface during accelerated corrosion tests in the climate chamber, t = 50 °C, RH 98%, recurrent (once in 10 days) immersion of a specimen into 0.1 M NaCl solution in accordance with the requirements of the Volvo VICT standard [46]: 1—unmodified copper; 2—copper modified by AS (15 molecular layers); 3—copper modified by VS (6 molecular layers); 4—copper modified by the [AS + VS] mixture (5 molecular layers).

**Figure 26 polymers-16-03066-f026:**
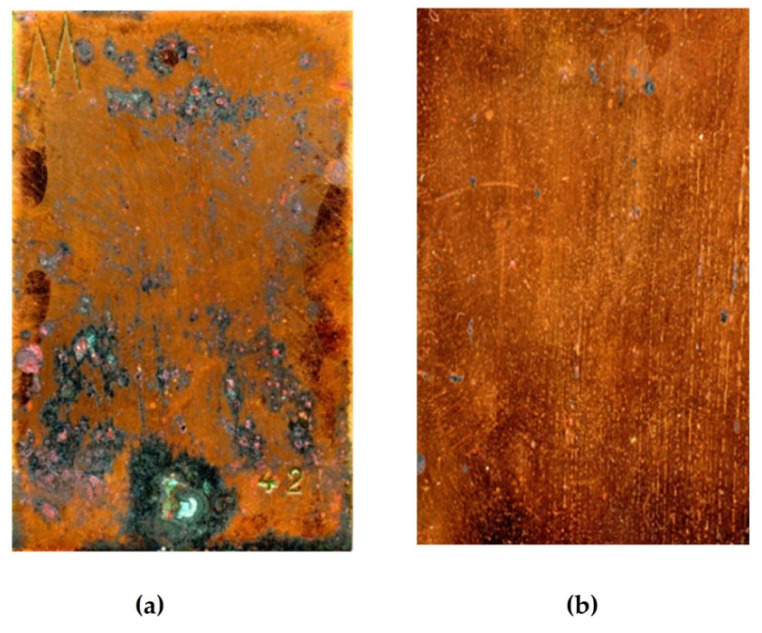
Appearance of specimens after 45 days of tests carried out according to the Volvo-VICT standard: climate chamber, t = 50 °C, RH 98%, recurrent (once in 10 days) immesrion of a specimen into 0.1 M NaCl solution. (**a**) unmodified copper; (**b**) copper modified with the [AS + VS] mixture (5 molecular layers).

**Figure 27 polymers-16-03066-f027:**
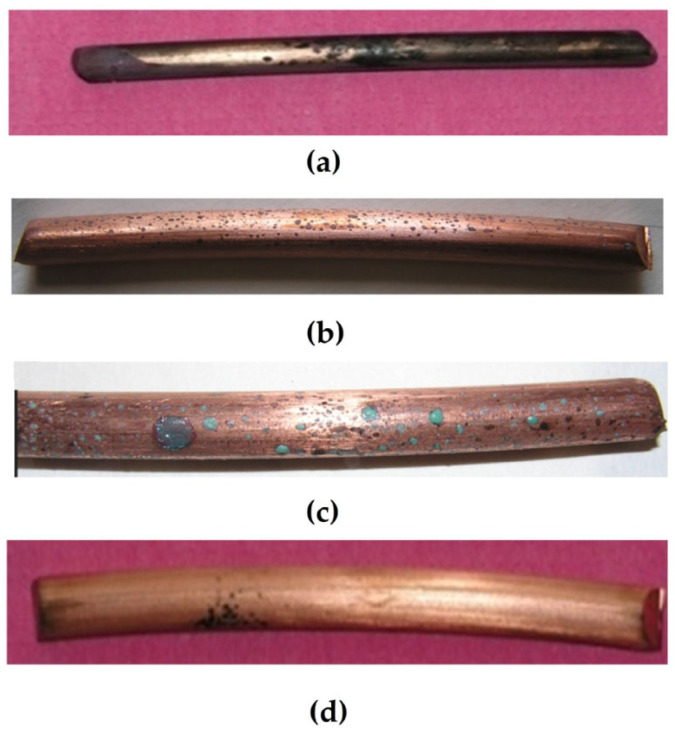
Change in the appearance of specimens after 35 days of accelerated corrosion tests: (**a**) unmodified copper; (**b**) copper pre-modified with 1% aqueous VS solution; (**c**) copper pre-modified with 1% aqueous AS solution; (**d**) copper pre-modified with 1% aqueous solution of the [1% VS + 1% AS] mixture. Climate chamber, t = 39 °C, RH 98%, recurrent immersion into NaCl solution.

**Figure 28 polymers-16-03066-f028:**
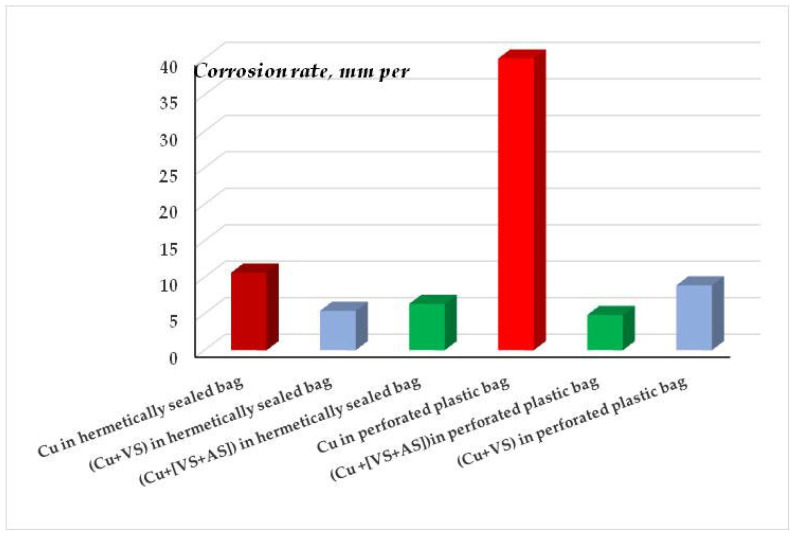
Corrosion rates of copper rod specimens packed in polyethylene bags (sealed or perforated) after 35 days of accelerated corrosion tests. Climate chamber, t = 39 °C, RH 98%, recurrent immersion into NaCl solution.

**Figure 29 polymers-16-03066-f029:**
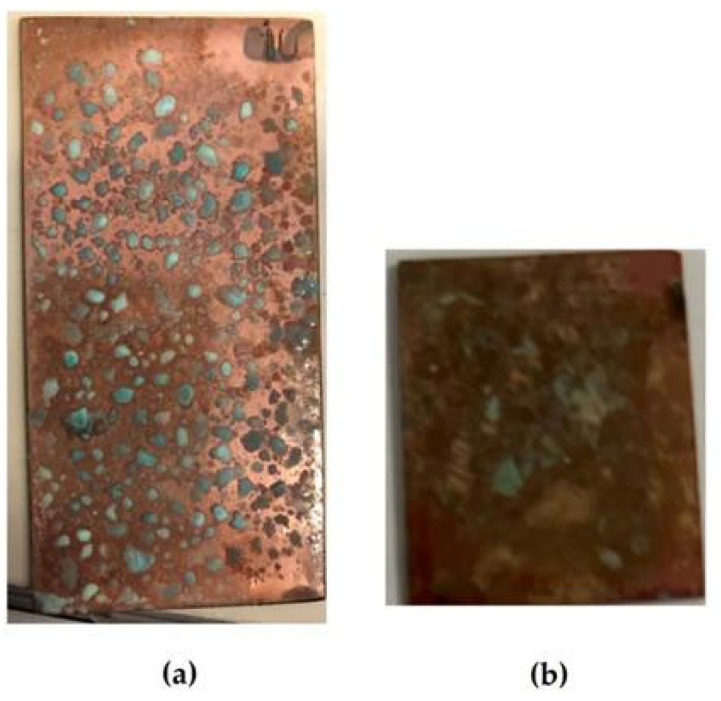
Surface of unmodified copper specimens after testing in a salt spray chamber: (**a**) 1 day (RH 96%, t = 25 °C, 60 min NaCl spraying); (**b**) 8 days (RH 96%, t = 25 °C, 6.5 h NaCl spraying).

**Figure 30 polymers-16-03066-f030:**
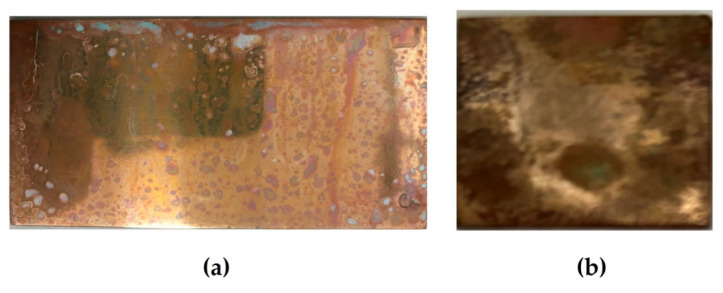
Surface of unmodified copper specimens after testing in a salt spray chamber: (**a**) 1 day (RH 96%, t = 25 °C, 60 min NaCl spraying); (**b**) 8 days (RH 96%, t = 25 °C, 6.5 h NaCl spraying). Droplets of the copper chloride-hydroxide solution were removed from the surface of the specimens by means of filter paper.

**Figure 31 polymers-16-03066-f031:**
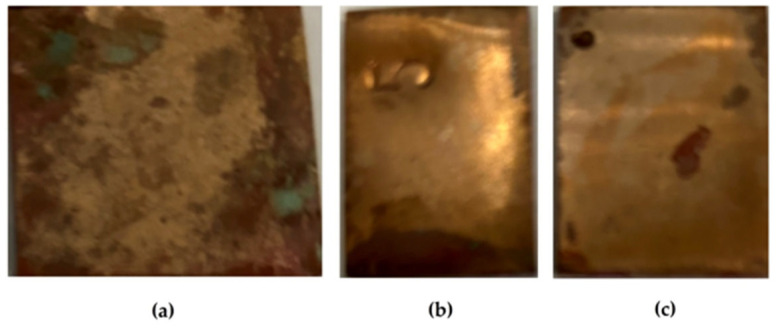
Surface of copper specimens modified with organosilanes after 8 days of testing in a salt spray chamber (RH 96%, t = 25 °C, NaCl spraying for 6.5 h). Droplets of the copper chloride-hydroxide solution were removed from the surface of the specimens using filter paper. (**a**) modification with an AS solution; (**b**) modification with a VS solution; (**c**) modification with a solution of the [VS + AS] mixture.

**Figure 32 polymers-16-03066-f032:**
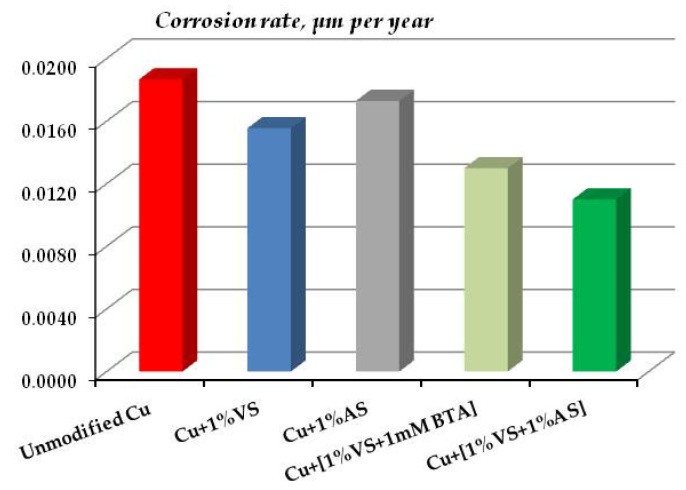
Corrosion rate of copper coated with organosilicon films in accelerated corrosion tests. Salt spray chamber. The test duration was 8 days, including NaCl spraying for 5 h. RH 96%, t = 35 °C.

**Figure 33 polymers-16-03066-f033:**
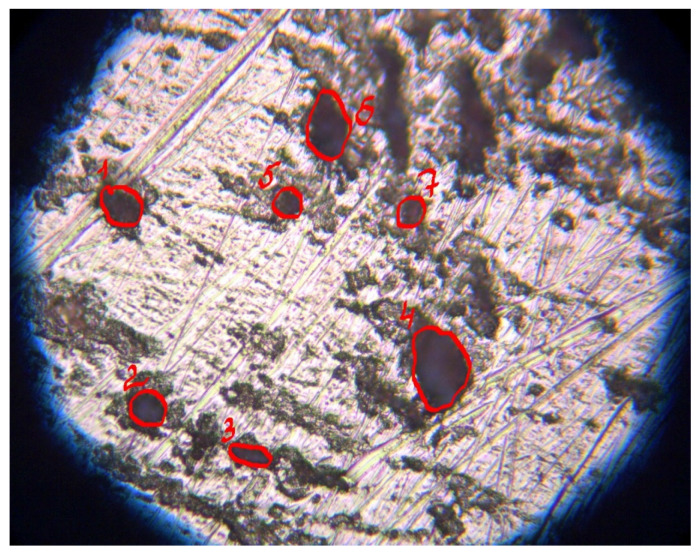
Appearance of a copper specimen after 8 days of accelerated corrosion testing in a salt spray chamber (RH 96%, t = 35 °C, NaCl spraying for 6.5 h) and removal of corrosion products from the surface. Highlighted in red are "deep” corrosion defects clearly visible on the surface even after removal of metal corrosion products (i.e. after chemical etching of the surface).

**Figure 34 polymers-16-03066-f034:**
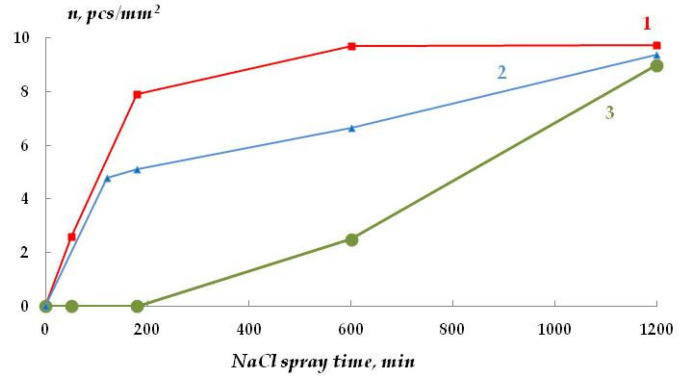
Variation in the density of surface coverage with corrosion defects vs. the time of spraying with sodium chloride. Accelerated corrosion tests of copper in a salt spray chamber. 1—unmodified copper; 2—copper modified with VS; 3—copper modified with a solution of the mixture [VS + AS].

**Figure 35 polymers-16-03066-f035:**
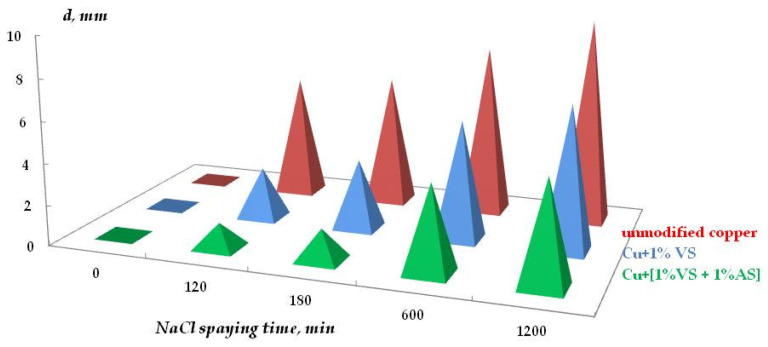
Variation in the geometric parameters (diameter) of a corrosion defect (pit) from time of NaCl spraying. Accelerated corrosion tests of copper in a salt spray chamber.

**Figure 36 polymers-16-03066-f036:**
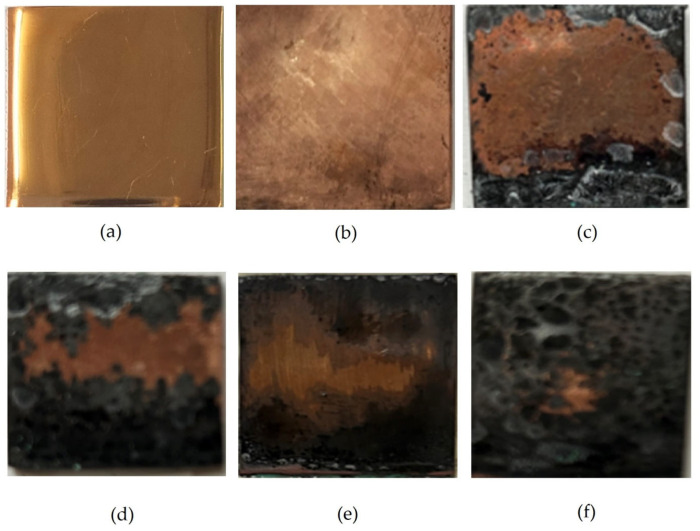
Change in the appearance of a copper specimen during accelerated corrosion tests in a climate chamber: (**a**) initial specimen; (**b**) specimen after 3 h of tests; (**c**) specimen after 1 day of tests; (**d**) specimen after 2 days of tests; (**e**) specimen after 7 days of tests; (**f**) specimen after 32 days of tests. Climate chamber, RH 95%, t = 25 °C.

**Figure 37 polymers-16-03066-f037:**
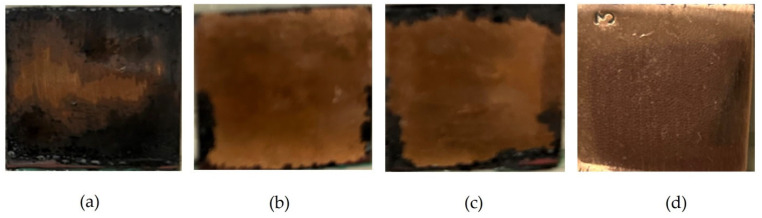
Appearance of copper specimens after 7 days of accelerated corrosion tests in a climate chamber: (**a**) unmodified copper; (**b**) copper pre-modified with a vs. solution; (**c**) copper pre-modified with an AS solution; (**d**) copper pre-modified with a [VS + AS] mixture solution. Climate chamber, RH 95%, t = 25 °C.

**Figure 38 polymers-16-03066-f038:**
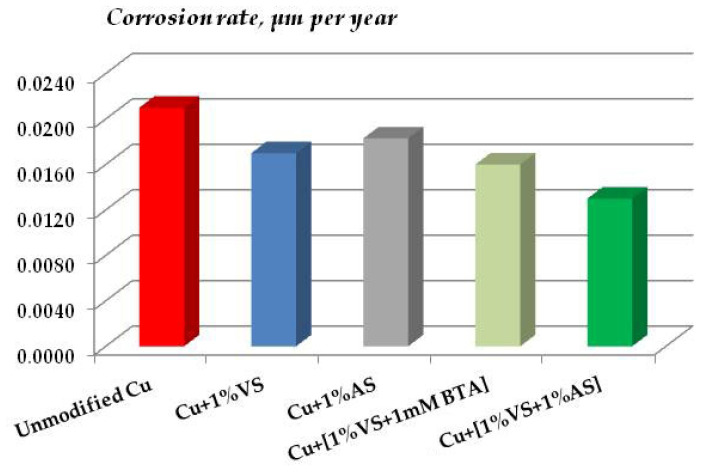
Corrosion rates of copper coated with organosilicon films in accelerated corrosion tests. Climate chamber, RH 95%, t = 60 °C. Test duration: 32 days.

**Figure 39 polymers-16-03066-f039:**
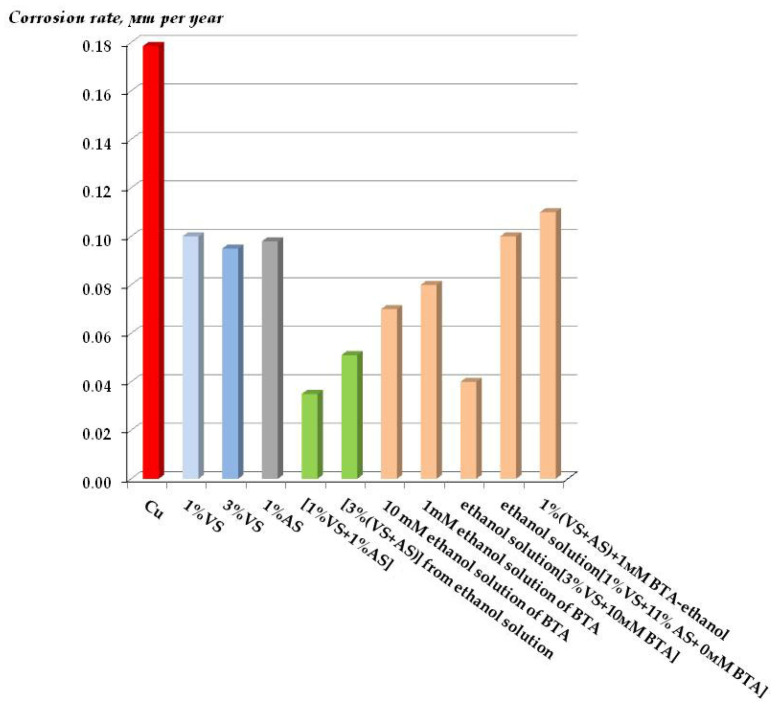
Results of outdoor corrosion tests of copper modified with organosilanes in an urban industrial atmosphere. Moscow Corrosion Test Station, Moscow, Russian Federation. Test duration: 1 year.

**Figure 40 polymers-16-03066-f040:**
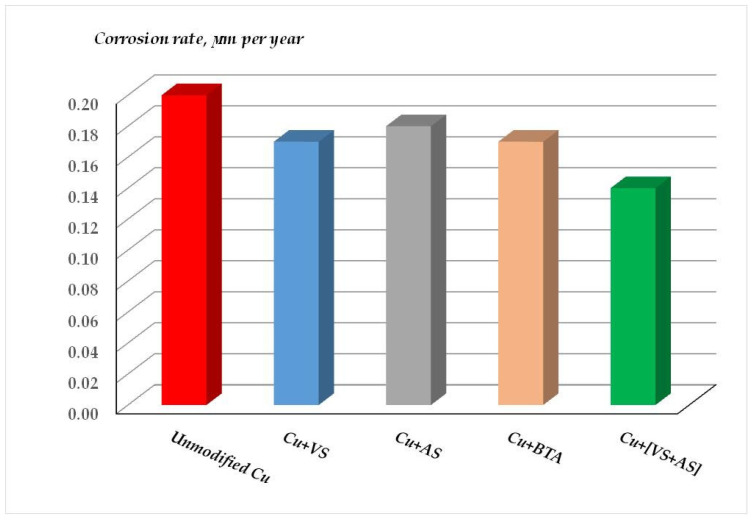
Results of outdoor corrosion tests of copper modified with organosilanes in an сoastal moderately cold atmosphere. Northern corrosion test station, Barents Sea coast, Murmansk region, Russian Federation. Test duration: 1 year.

**Table 1 polymers-16-03066-t001:** Compositions of the modifying solutions.

No.	Modifier	Solvent	Compositions of the Modifying Solutions
1	VS	water	1% VS
2	AS	water	1% AS
3	[VS + AS] mixture	water	1% VS +1% AS
4	BTA	Ethyl alcohol	10 mM BTA
5	[BTA + VS] mixture	Ethyl alcohol	10 mM BTA +1% VS

**Table 2 polymers-16-03066-t002:** Modes of copper rod testing.

Mode	Description
1	Air temperature +40 ± 1 °C. Relative humidity 98 ± 2%. Chloride aerosol deposition rate 300 mg/m^2^ per day. Duration 20 h.
2	Cyclic air temperature variations: maximum +20 ± 3 °C, duration 4 h, minimum −15 ± 1 °C, duration 4 h. Temperature transition time 30 min. 100 cycles—800 h.
3	Air temperature + 40 ± 1 °C. Relative humidity 98 ± 2%. Once in 3 days, specimens were immersed in 1% NaCl solution at +40 ± 1 °C for 1 h. Duration 20 h.

**Table 3 polymers-16-03066-t003:** Thickness of films formed on copper surface after modification with organosilane solutions.

Modifier	Film Thickness, µm
VS, 1% aqueous solution	0.168
AS, 1% aqueous solution	0.302
Mixture [1% VS +1% AS], aqueous solution	0.899

**Table 4 polymers-16-03066-t004:** Pitting potentials of copper specimens modified with organosilane-based formulations.

System	E_pt_	ΔE_pt_ = E_pt-mod_ − E_pt-bgd_	Effect on Pitting
Cu	0.933	0	-
Cu + 1% VS	1.104	0.071	Inhibition
Cu + [1% AS + 1% VS]	1.079	0.146	Noticeable inhibition
Cu + 1% AS	0.913	−0.02	Weak activation
Cu + [1% VS + 10 mM BTA]	0.935	0.102	Inhibition
Cu + 10 mM BTA	0.813	−0.012	Very weak activation

**Table 5 polymers-16-03066-t005:** Corrosion rate of copper in the presence of vinylsiloxane nanolayers on the surface.

Vinylsiloxane Layer Thickness, Molecular Layers	Copper Corrosion Rate, % of Corroded Surface/h
0.00	16.5
0.87	15.3
1.52	13.7
3.81	3.9
19.31	6.7

**Table 6 polymers-16-03066-t006:** Mass losses of specimens in a hermetically sealed package after 35 days of accelerated corrosion tests (mode 1).

System	Δm, g
Bag, cleaned specimens	0.0014
Bag, as-received specimens	0.0009
Bag, specimens with GKZh coating	0.0004
Bag, specimens with Libro coating	0.0009
Bag, specimens with VS coating	0.0005
Bag, specimens with AS coating	0.0005
Bag, specimens with [VS + AS] coating	0.0002
Bag, specimens with 118-BBK coating	0.0011
Bag, specimens with 118-G coating	0.0007
Bag, specimens with 118 BTA coating	0.0008

**Table 7 polymers-16-03066-t007:** Mass losses of specimens in an airtight package after accelerated climate tests (mode 1).

System	Δm, g
Cleaned specimens	0.0033
Cleaned specimens, NaCl spraying	0.0153
As-received specimens, NaCl spraying	0.0127
Specimens with GKZh coating, NaCl spraying	0.0034
Specimens with Libro coating, NaCl spraying	0.0155
Cleaned specimens, immersion in NaCl	0.0378
As-received specimens, immersion in NaCl	0.0348
Specimens with GKZh coating, immersion in NaCl	0.0078
Specimens with Libro coating, immersion in NaCl	0.0420
Perforated bag, cleaned specimens	0.0165
Perforated bag, as-received specimens	0.0024
Perforated bag, specimens with GKZh coating	0
Perforated bag, specimens with Libro coating	0.0001
Perforated bag, specimens with VS coating	0.0008
Perforated bag, specimens with AS coating	0.00024
Perforated bag, specimens with [VS + AS] coating	0

**Table 8 polymers-16-03066-t008:** Effect of organosilicon surface layers on the atmospheric corrosion of copper in an urban industrial atmosphere. Moscow Corrosion Test Station, Moscow, RF. Duration of tests: 1 year.

Systems/Modifying Solutions	Corrosion Inhibition Coefficient, γ
Unmodified copper	1.0
1%VS	1.8
3%VS	1.9
1%AS	1.8
[1% VS + 1% AS]	5.1
ethanol solution of [3% VS + 1% AS]	3.5
10 mM ethanol solution of BTA	2.6
1mM ethanol solution of BTA	2.2
ethanol solution of [3% VS + 10 mM BTA]	4.5
ethanol solution of [1% VS + 11% AS + 0 mM BTA]	1.8
1% (BC + AC) + 1 mM BTA, ethanol	1.6

**Table 9 polymers-16-03066-t009:** Effect of organosilicon surface layers on the atmospheric corrosion of copper in an urban industrial atmosphere. Northern corrosion test station, Barents Sea coast, Murmansk region, Russian Federation. Duration of tests: 1 year.

Systems/Modifying Solutions	Corrosion Inhibition Coefficient, γ
Unmodified copper	1.0
Cu + VS	1.1
Cu + AS	1.2
Cu + BTA	1.1
Cu + [VS + AS]	1.5

## Data Availability

Data are contained within the article.

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
