# Peer review of "Effect of Surface Polymeric Organosilicon Nanolayers on the Electrochemical and Corrosion Behavior of Copper"

_polymers, 2024, doi:10.3390/polym16213066_

Round 1
Reviewer 1 Report
Comments and Suggestions for Authors
This paper seems to be extensive and important for addressing copper corrosion. Here are some comments and recommendations.
1. Alignment issues in text especially on page 5 of 37 makes it difficult to read and understand what author is implying.
2. The contact angles measurements show how the surface has changed. Superhydrophobic copper surfaces will have better properties. If Silane is not creating hydrophobic surface, then enhancement in corrosion resistance needs to be well understood.
3. The surface morphological analysis, before and after corrosion tests has been conducted to evaluate the corrosion performance. It’s also important to understand the kinetics of formation of the pits on surface and how pits are growing if they are.
4. The surface composition of the functionalized copper coupons can be easily done by X-ray photoelectron spectroscopy (XPS) or Energy Dispersive X-ray Spectroscopy (EDS). That will be important to understand the formation of patina at lowest percentages.
5. Stability and durability of the Silane treated copper surface against corrosion can then be established.
Comments on the Quality of English LanguageMinor sentence structure are needed and some repetition has to be cleaned off.
Author Response
Comments 1. Alignment issues in text especially on page 5 of 37 makes it difficult to read and understand what author is implying.
Response 1. It is not very clear what "Alignment issues in text" are meant. We decided that it was about the formula, since all the rest of the text on page 5 is aligned, and corrected all the formulas on page 5 of 37.
Comments 2: The contact angles measurements show how the surface has changed. Superhydrophobic copper surfaces will have better properties. If Silane is not creating hydrophobic surface, then enhancement in corrosion resistance needs to be well understood.
Response 2: The objective of this study was not to evaluate the hydrophobicity or hydrophilicity of the copper surface, so wetting angle measurements were not performed. Although we (the authors) understand that the value of the wetting angle characterizes the change in both the geometry and thermodynamic characteristics of the metal surface, which was noted in the literature (see the link below [1]). The objective of this work was to determine the effect of siloxane layers on copper corrosion, and the value of the contact angle only indirectly indicates the tendency (or non-tendency) of the metal to corrosive destruction. The literature does not indicate a clear correlation between the degree of surface hydrophobization and its corrosion resistance, although from general considerations it is obvious that the more hydrophobic the surface, the better it resists corrosion, especially in atmospheric conditions, since the hydrophobic coating creates a barrier for the delivery of corrosive agents (water and oxygen) to the metal surface. Superhydrophobic coatings certainly reduce metal corrosion (see, forexample, the references below [2,3]). However, for superhydrophobization of the surface, it is not enough to apply a siloxane coating, but it is necessary to carry out special surface preparation - texturing (most often laser texturing) followed by application of a superhydrophobic surface film (obtained on the basis of functional (surface-active) compounds with a long saturated hydrocarbon radical – carboxylic acids, phosphonic acids, etc., including organosilanes). Despite the fact that we did not determine the value of the wetting angle of the copper surface, it is known from the literature (see the references below [4,5]), that treatment of the metal surface with organosilanes that do not carry a long organic radical leads not to superhydrophobization, but to ordinary hydrophobization(i.е. θ>900) of the surface, especially acting in a mixture with carboxylic acids (for example, with stearic [5]) and to corrosion inhibition. [5].
- Boinovich, A. Emelyanenko. Characterizing the Physicochemical Processes at the Interface through Evolution of the Axisymmetric Droplet Shaoe Parameters. in Advances in Contact Angle, Wettability and Adhesion.V.3. K.L. Mittal Ed. 2018. John Wiley & Sons, Inc. Hoboken, NJ, USA. P. 99-130
- M Semiletov., A.A.Kudelina, Yu.I. Kuznetsov. Superhydrophobization of the surface of aluminum alloy with organic acids to protect against atmospheric corrosion. Russian Journal of Physical Chemistry. A. 2023.V. 97. No. 2. P. 397-404.
- M. Semiletov, A.A. Kudelina, Yu.I. Kuznetsov. A superhydrophobic coating and corrosion inhibitors as a combined method for the protection of aluminum alloys in chloride solutions. International Journal of Corrosion and Scale Inhibition. 2023. V. 12. No. 3. P. 1321-1335.
- Ferrari, F. Cirisano. Hydrophobicity and Superhydrophobicity Fouling Prevention in Sea Environment. In Advances in Contact Angle, Wettability and Adhesion. V.3. K.L. Mittal, Ed. 2018. John Wiley & Sons, Inc. Hoboken, NJ, USA. P. 241-298.
- I. Kuznetsov, A.M. Semiletov, A.A. Chirkunov, I.A. Arkhipushkin, L.P. Kazanskii, N.P. Andreeva. Protecting aluminum from atmospheric corrosion via surface hydrophobization with stearic acid and trialkoxysilanes. Russian Journal of Physical Chemistry A. 2018. V. 92. No. 4. P. 621-629.
Comments 3: The surface morphological analysis, before and after corrosion tests has been conducted to evaluate the corrosion performance. It’s also important to understand the kinetics of formation of the pits on surface and how pits are growing if they are.
Response 3: We investigated the kinetics of development of both uniform and localized (pitting) corrosion of copper: Figures 30 and 32 show the kinetics of uniform corrosion in terms of metal mass loss and surface area change due to metal corrosion products. Figure 34 shows the kinetics of copper pitting corrosion development in terms of change in the specific area of the sample occupied by pits. Figure 43 shows the kinetics of change in the surface filling density by pits, i.e. change (due to occurrence) in the number of pits per unit area of the sample. Figure 44 shows the kinetics of change in the geometric dimensions of an individual pit.
Notes:
Due to the reviever’s comments, we made some changes:
- Now Figure 30 became Figure 21
- Now Figure 32 became Figure 23
- Now Figure 34 became Figure 25
- Now Figure 43 became Figure 34
- Now Figure 44 became Figure 35
Comments 4: The surface composition of the functionalized copper coupons can be easily done by X-ray photoelectron spectroscopy (XPS) or Energy Dispersive X-ray Spectroscopy (EDS). That will be important to understand the formation of patina at lowest percentages.
Response 4: We do not quite understand the expression "to understand the formation of patina at the lowest percentages." Apparently, the respected reviewer means the use of the specified methods to assess the degree of corrosion. Unfortunately, we were unable to use these methods in this way because we did not have the opportunity to apply them “in situ” during the corrosion experiment. However, in our previous works (see the link below [1]) we used exactly these methods to characterize the surface organosilane layer on mild steel and found that a polymer/oligomeric siloxane layer (adjacent molecules on the surface are connected by bridging ≡Si-О-Si≡ bonds) bound to the metal (by Me-O-Si≡ bonds) is formed on the metal surface. And when using mixtures of organosilane with benzotriazole, additional crosslinking of molecular chains is observed due to the formation of silazane (=Si-N=) bonds.
- Gladkikh, Yu. Makarychev, M. Maleeva, M. Petrunin, L. Maksaeva, A. Rybkina, A. Marshakov, Yu. Kuznetsov. Synthesis of thin organic layers containing silane coupling agents and azole on the surface of mild steel. Synergism of inhibitors for corrosion protection of underground pipelines. Progress in Organic Coatings. 2019. V.132. P.481-489.
Comments 5: Stability and durability of the Silane treated copper surface against corrosion can then be established.
Response 5: Unfortunately, within the framework of this work, the assessment of stability and durability was not carried out. However, earlier, in the previous work (see the link below [1]), using Fourier-transformed IR spectroscopy (IR spectra were taken before and after corrosion tests) it was shown that the surface siloxane layer is preserved on the zinc surface for several days under conditions of an intensive corrosion process - in a 0.1 M NaCl solution. And there is every reason to expect that on copper, even in a corrosive solution, it will also be stable (for several days), and in milder atmospheric conditions it can be assumed that the life time of the layer will be several weeks, and even months.
- Petrunin, L. Maksaeva, T. Yurasova. Organosilicon self-assembled surface nanolayers on zinc — formation and their influence on the electrochemical and corrosion zinc ongoing. Materials. 2023, 16, 6045.
Reviewer 2 Report
Comments and Suggestions for Authors
COMMENTS ON MANUSCRIPT Polymers-3221792
TITLE: Effect of surface polymeric organosilicon nanolayers on the electrochemical and corrosion behavior of copper
COMMENTS
The work reports on the corrosion inhibition effect of polymeric organosilicon nanolayers on copper surface in 0.1 M NaCl solution and atmospheric conditions. The corrosion inhibition effect was investigated electrochemical technique complemented by surface morphological characterization of the corroded copper samples without and with the polymeric organosilicon nanolayers with SEM and AFM. The work is of interest in the field of materials and corrosion. Overall assessment shows that the manuscript is well written, and conclusions drawn are supported by the data. However, a major revision and reorganization of the manuscript is required before the manuscript will be considered for publication in Polymers. The issues to be addressed by the authors are appended below.
(1) In the abstract, the unit of measurement should be added to the thickness of the nanolayers.
(2) The authors should point out the various ways adopted to control copper corrosion before concluding that there is no reliable way to prevent copper corrosion to date.
(3) In line 59, the numbers in the alkoxysilanes general formula should be properly subscripted.
(4) The figures are too many. It is recommended that some of them should be merged instead of standing alone.
(5) In Figures 15 and 16, it is not known which one is (1) and (2). The curves should be properly labelled.
(6) It is noted that only one electrochemical technique (potentiodynamic polarization) was used. For comparison purposes another electrochemical technique preferably electrochemical impedance spectroscopy (EIS) should be added.
(7) The conclusion should be rewritten to make meaning.
Comments on the Quality of English Language
The English language is okay though need minor revision.
Author Response
Comments 1. In the abstract, the unit of measurement should be added to the thickness of the nanolayers.
Response 1. The abstract of the article does indeed provide the layer thickness, which can be expressed in units of thickness (length). The abstract of the article does indeed provide the layer thickness, which can be expressed in units of thickness (length). Unfortunately, direct measurements of the layer thickness were not carried out in the work, so we did not provide them in the text of the article. Using quartz nanoweighing, we determined the mass of the layer, which can be easily converted into the volume of the layer (formula (3) in the article) and then, knowing the area, into the thickness (formula (4) in the article.) But such a recalculation is correct only if the layer density is known. It can be expected that the density of the polycondensed siloxane layer will be almost the same as that of organosilanes. However, in very thin layers on the surface, the value of the layer density may differ from the bulk density, so it is hardly possible to correctly calculate the layer thickness from gravimetric data (mass). On the other hand, processing of adsorption data using commonly used adsorption approaches (for example, in terms of Langmuir or BET isotherms) allows one to calculate the so-called monolayer capacity, i.e. the number of adsorbate molecules required for dense monolayer filling of the surface. Knowing this value and determining the amount of adsorbed substance, it is easy to determine the number of molecular layers of the adsorbate on the surface: if the amount of adsorbed substance coincides with the capacity of the monolayer, then the layer thickness is 1 monolayer; if it exceeds twice, then 2 molecular layers; three times - 3 molecular layers, etc. Therefore, we consider it correct to use "molecular layers" as a unit of measurement of the surface layer thickness. In our article, in the abstract, the given layer thickness of 3.8 molecular layers is defined as shown above.
Comments 2: The authors should point out the various ways adopted to control copper corrosion before concluding that there is no reliable way to prevent copper corrosion to date.
Response 2: We have added the following text to the article - end of line 57, immediately after "...Humphry Davy [17]" and changed the list of references by adding 8 references after reference [17]:
«who developed the principles of electrochemical protection of copper sheets of the hull of sea ships [17]. Since then, various methods have been proposed to protect copper from corrosion, for example, sometimes or to preserve a high-quality finish, protective metallic coatings of one or more of the following metals are used: tin, lead, nickel, silver, chromium, rhodium, gold [9]. In some conditions, corrosion inhibitors are recommended as a protective measure. Nitrogen-containing compounds, such as those with an azole group in the molecule, such as benzotriazole, have proven to be the most effective for copper. Sulfur-containing organic corrosion inhibitors, such as sodium diethyldithiocarbamate, are also effective [9]. But the most applicable method of protection is the use of organic coatings: for example, in some aggressive soils - bitumen or polymeric [9], and in less severe conditions - paint and varnish [5,9] or for copper and brass - transparent varnish [5]. In addition, the literature provides information on the protection of copper from corrosion by organosilicon coatings obtained by treating the copper surface with organosilanes [18-24], which often were used in a mixture with nitrogen- [22,23], sulfur- [18,21,24] and phosphorus- [22] containing compounds (inhibitors) or with the addition(filled) of graphene oxide nanoparticles [20]».
And further on line 56 before "there is no,..”.
«However, despite many years of research efforts aimed at developing means of anti-corrosion protection for copper…”
Notes:
Due to the reviever’s comments, we made some changes:
- Now reference 18 became reference 25.
- Now reference 19 became reference 26.
- Now reference 20 became reference 27.
- Now reference 21 became reference 28.
- Now reference 22 became reference 29.
- Now reference 23 became reference 30.
- Now reference 24 became reference 31.
Comments 3: In line 59, the numbers in the alkoxysilanes general formula should be properly subscripted.
Response 3: In the general formula of alkoxysilanes on line 59 the numbers are converted to lower case
Comments 4: The figures are too many. It is recommended that some of them should be merged instead of standing alone.
Response 4: We have reduced the total number of figures in the article by combining Figures.
Notes:
Due to the reviever’s comments, we made some changes:
- Now Figure 4 became Figure 4 (a).
- Now Figure 5 became Figure 4 (b).
- Now Figure 11 became Figure 10 (a).
- Now Figure 12 became Figure 10 (b).
- Now Figure 13 became Figure 10 (c).
- Now Figure 14 became Figure 10 (d).
- Now Figure 15 became Figure 11 (a).
- Now Figure 16 became Figure 11 (b).
- Now Figure 17 became Figure 12, curve 1.
- Now Figure 18 became Figure 12, curve 2.
- Now Figure 19 became Figure 12, curve 3.
- Now Figure 20 became Figure 12, curve 4.
- Now Figure 21 became Figure 12, curve 5
Comments 5: In Figures 15 and 16, it is not known which one is (1) and (2). The curves should be properly labelled.
Response 5: We have corrected Figure 15 and 16 by adding curve labels and combined them into one Figure.
Notes:
Due to the reviever’s comments, we made some changes:
- Now Figure 15 became Figure 11 (a).
- Now Figure 16 became Figure 11 (b).
Comments 6: It is noted that only one electrochemical technique (potentiodynamic polarization) was used. For comparison purposes another electrochemical technique preferably electrochemical impedance spectroscopy (EIS) should be added.
Response 6: We used the potentiodynamic method to determine the effect of surface layers on local metal dissolution. The obtained polarization curves allow us to determine the pitting potential (Ept) (the observed characteristic bend in the curve), i.e. the potential above which an intensive process of local (pitting) metal dissolution occurs. When using corrosion inhibitors that form surface layers, the effect of the inhibitor (layer) on the pitting process can be determined by the magnitude of the change in the pitting potential: a positive shift in Ept indicates inhibition of pitting dissolution, while a negative shift indicates activation of this process. The use of electrochemical impedance spectroscopy (EIS) in these conditions has limited capabilities, since it allows us to determine the value of polarization resistance (Rp), which characterizes the process of uniform metal dissolution. This case is less dangerous, and therefore less interesting for study. We would like to note that we are currently conducting research into the characteristics of surface siloxane layers obtained under different conditions using the EIS method.
Comments 7: The conclusion should be rewritten to make meaning.
Response 7: We have changed/edited the conclusion and inserted a new version of the conclusion into the revised manuscript (line 981).
- It has been established that preliminary modification of the copper surface with aqueous solutions of individual organosilanes (both vinyl- and amino-containing) leads to the formation of nanosized polymer surface layers with a thickness not exceeding 300 nm. Treatment of the copper surface with an aqueous solution of a mixture of vinyl- and amino-containing organosilanes leads to the formation of "thicker" (~1 micron thick) and uniformly (by thickness) distributed siloxane layers over the surface.
- Copper corrosion in a weakly acidic chloride-containing electrolyte was studied. The composition of the layer of corrosion products formed on the surface during testing was determined. It was shown that a copper chloride film was formed on the surface during the first 5-7 hours of testing. An increase in the duration of holding the samples in the solution led to the growth of a two-layer film of corrosion products on the surface: the first layer was an ordered film consisting of 87% Cu2O and 13% CuCl, the second layer was a loose porous film of similar composition.
- The effect of vinylsiloxane self-assembled nanolayers on copper corrosion was studied and it was shown that the surface organosilicon layers formed on copper during modification effectively inhibit anodic dissolution and corrosion (including local) of copper in aggressive electrolytes, as well as in artificial and natural atmospheres. It was found that the protective effect of the nanolayer depends on its thickness. The maximum protection efficiency was observed at a thickness of 3.8 molecular layers, at which the densest layer is formed, hindering the adsorption of chloride ions and significantly reducing the rate of their interaction with surface copper atoms.
Reviewer 3 Report
Comments and Suggestions for Authors
The authors' work addresses organosilane coatings for the corrosion protection of copper surfaces. The study is comprehensive and impresses with the various lab analyses and field tests that were carried out. The results obtained appear to be interesting for practical applications. However, the corrosion protection effect of organosilanes on copper has already been examined in several other studies, hence the novelty aspect of this study remains unclear. Furthermore, the literature review lacks a summary of the numerous studies and theirs findings on coatings for corrosion protection of copper. In particular, studies on organosilane coatings for copper protection are missing. A few studies (Ref: 18..21) on the protection of steel and other metals are cited, but relevant papers for copper protection should be included. The conclusion should be improved, which is currently a summary and should place the results of this study in a broader context. In addition, the manuscript contains shortcomings in terms of clarity and presentation of results. Overall, the novelty aspect of the study should be addressed, and the manuscript should be thoroughly revised, taking into account the points mentioned above and the minor points mentioned below.
Minor points:
- The description of the methodology on page 6,7 is difficult to understand. Perhaps a graphic visualization with a flow chart could help here
- Typo on line 294
- Line 394: how were the mentioned 0.8-0.9 um thickness of islands determined
- Figure 15: orange and blue graphs are not labelled
- Line 406..414: Explanations for Figure 15 are difficult to understand from the graphs
- Figure 16: orange and blue graphs are not labelled
- Figures 17-21 could be integrated into a panel figure. Then it would be more compact and easier to compare.
- Line 469: how was Ept determined?
- Table 4: It would be desirable to compare the Ept obtained with other studies
- Line 655: Capital T is usually used for the absolute temperature
- Line 759: Weird sentence. Suggestion “One can see from Figure 40 that surface modification with 759 solutions of individual silanes leads to the inhibition of copper corrosion, as observed in 8 days of accelerated corrosion tests in a salt spray chamber”
- Line 853: Sentence uncomplete.
- Line 879: Typo ~1 m
Comments on the Quality of English Language
Author Response
Comments 1. In particular the corrosion protection effect of organosilanes on copper has already been examined in several other studies, hence the novelty aspect of this study remains unclear.
Response 1. We have changed the introduction (line 57) and reference list (line 105-1067) and added information on copper corrosion protection by organosilane layers. References 18-24 have been added to the reference list. The following phrase has been added to the introduction:
«In addition, the literature provides information on the protection of copper from corrosion by organosilicon coatings obtained by treating the copper surface with organosilanes [18-24], which often were used in a mixture with nitrogen-[22,23], sulfur-[18,21,24] and phosphorus-[22] containing compounds (inhibitors) or with the addition(filled) of graphene oxide nanoparticles [20]».
It was especially noted that most of the cases described in the literature on the use of organosilanes for anticorrosive protection of copper (organosilanes) concern mixtures of organosilanes with nitrogen- or sulfur-containing compounds, which can act as inhibitors of copper corrosion, or mixtures of organosilanes with nanofillers of surface composites, such as nanoparticles, nanoplates, and nanotubes made of functionalized graphene oxide. In our work,organosilanes are used to protect copper from corrosion without any additives. This can be considered the novelty of this work.
Notes:
Due to the reviever’s comments, we made some changes:
- Now reference 18 became reference 25.
- Now reference 19 became reference 26.
- Now reference 20 became reference 27.
- Now reference 21 became reference 28.
- Now reference 22 became reference 29.
- Now reference 23 became reference 30.
- Now reference 24 became reference 31.
Comments 2. The literature review lacks a summary of the numerous studies and theirs findings on coatings for corrosion protection of copper.
Response 2. We have changed the introduction (line 57) and added the following insert about protective coatings on copper:
«But the most applicable method of protection is the use of organic coatings: for example, in some aggressive soils - bitumen or polymeric [9], and in less severe conditions - paint and varnish [5,9] or for copper and brass - transparent varnish [5]».
Comments 3. The conclusion should be improved.
Response 3. We have rewritten the Conclusions (line 981) and inserted the revised version of conclusions into the text of the article:
- It has been established that preliminary modification of the copper surface with aqueous solutions of individual organosilanes (both vinyl- and amino-containing) leads to the formation of nanosized polymer surface layers with a thickness not exceeding 300 nm. Treatment of the copper surface with an aqueous solution of a mixture of vinyl- and amino-containing organosilanes leads to the formation of "thicker" (~1 micron thick) and uniformly (by thickness) distributed siloxane layers over the surface.
- Copper corrosion in a weakly acidic chloride-containing electrolyte was studied. The composition of the layer of corrosion products formed on the surface during testing was determined. It was shown that a copper chloride film was formed on the surface during the first 5-7 hours of testing. An increase in the duration of holding the samples in the solution led to the growth of a two-layer film of corrosion products on the surface: the first layer was an ordered film consisting of 87% Cu2O and 13% CuCl, the second layer was a loose porous film of similar composition.
- The effect of vinylsiloxane self-assembled nanolayers on copper corrosion was studied and it was shown that the surface organosilicon layers formed on copper during modification effectively inhibit anodic dissolution and corrosion (including local) of copper in aggressive electrolytes, as well as in artificial and natural atmospheres. It was found that the protective effect of the nanolayer depends on its thickness. The maximum protection efficiency was observed at a thickness of 3.8 molecular layers, at which the densest layer is formed, hindering the adsorption of chloride ions and significantly reducing the rate of their interaction with surface copper atoms.
Minor points:
Comments 1. The description of the methodology on page 6,7 is difficult to understand. Perhaps a graphic visualization with a flow chart could help here.
Response 1. We apologize for the lengthy and apparently not very clear description of the corrosion testing process on pages 6,7. We conducted the tests in accordance with the description of the original Volvo Cycle (Volvo-VICT) production accelerated corrosion tests presented in the literature, and in the article we tried to describe the entire procedure in as much detail (step by step) as possible. As for the graphical visualization of the procedure, at first we did not think to create it. Now we have not been able to implement it, unfortunately, and provide it both in the article and in these answers.
Comments 2: Typo on line 294.
Response 2: We have corrected the typo on line 294:
“sweep rate of potential was 1mV/s” (Due to the reviever’s comments, we made some changes.
Now line 294 became line 333).
Comments 3: Line 394: how were the mentioned 0.8-0.9 um thickness of islands determined.
Response 3: The thicknesses of the "islands" mentioned on line 349 were not determined by a direct method.
A rather rough rough (approximate) estimate of the layer thickness by a method based on the observation of interference colours in reflected light, which are caused by double reflection and refraction of white light passing through a transparent film and reflecting off an opaque substrate. In this case, due to the difference in the path of the rays, interference occurs, and therefore thin transparent films appear colored in reflected light. Their color depends only on the thickness and refractive index:
Δ = 2nd sinα,
where Δ is the difference in the path of the rays, n is the refractive index of the film, α is the angle of reflection, d is the thickness of the film.
This method was proposed by Newton at the end of the 17th century. The accuracy of the film thickness estimate determined by this method depends on the value of the film refractive index. We were guided by reference values of organosilanes refractive indices and by literary estimates of the degree of change in the refractive index after polymerization and formation of a surface film.
Comments 4: Figure 15: orange and blue graphs are not labeled.
Response 4: We have made corrections: put numbers on the curves in Figure 15 and inserted the revised Figure into the article.
Notes:
Due to the reviever’s comments, we made some changes.
Now Figure 15 became Figure 11 (a).
Comments 5: Line 406,.414: Explanations for Figure 15 are difficult to understand from the graphs.
Response 5: When discussing figures 15, 16, we are talking about the fact that it is difficult to indirectly judge the processes occurring on the metal surface by the change in the corrosion potential. Thus, according to some ideas, the potential shifts to the negative side when the film covering the metal surface is an n-type semiconductor, and the potential shifts to the positive side when the film is a p-type semiconductor. Guided by these ideas, it is possible to estimate (approximately, of course) the composition of the corrosion products during tests in solution. Analysis of figures 15, 16 showed that in the initial period a layer of CuCl is formed on the surface of unmodified copper. Further holding of copper in the solution, accompanied by an increase in potential, may indicate the formation of a mixed oxide-chloride layer: Cu2O-CuCl. The presence of a silicon-organic layer on the surface first causes the growth of a Cu2O film, and only after a certain period of time (longer than for unmodified copper) is a potential increase observed, corresponding to the formation of a mixed oxide-chloride film. This may indicate that the siloxane layer hinders the adsorption of chloride ions on the surface and, as a consequence, can inhibit copper corrosion in chloride-containing solutions.
Notes:
Due to the reviever’s comments, we made some changes:
- Now Figure 15 became Figure 11 (a).
- Now Figure 16 became Figure 11 (b).
Comments 6: Figure 16: orange and blue graphs are not labeled.
Response 6: We have made corrections: put numbers on the curves in Figure 16 and inserted the revised figure into the article.
Notes:
Due to the reviever’s comments, we made some changes:
Now Figure 16 became Figure 11 (b).
Comments 7: Figures 17-21 could be integrated into a panel figure. Then it would be more compact and easier to compare.
Response 7: We have combined Figure 17-21 into one Figure and inserted the revised figure into the article.
Notes:
Due to the reviever’s comments, we made some changes:
- Now Figure 17 became Figure 12, curve 1.
- Now Figure 18 became Figure 12, curve 2.
- Now Figure 19 became Figure 12, curve 3.
- Now Figure 20 became Figure 12, curve 4.
- Now Figure 21 became Figure 12, curve 5.
Comments 8: Line 469: how was Ept determined?
Response 8: Ept is determined from the polarization curve. The value of Ept corresponds to the potential of a sharp break in the anodic curve (see references [41], [42] in the article).
Notes:
Due to the reviever’s comments, we made some changes:
- Now reference 41 became reference 48.
- Now reference 42 became reference 49.
Comments 9: Table 4: It would be desirable to compare the Ept obtained with other studies.
Response 9: The obtained value of Ept for unmodified copper is close to the values presented in the literature. And if the data on the pitting potentials of copper can be found in the literature, then it is difficult to find the values of Ept for copper coated with siloxane layers. Therefore, we can only talk about the values of Ept for unmodified copper, which are close to the values presented in the literature devoted to the corrosion behavior of copper.
Comments 10: Line 655: Capital T is usually used for the absolute temperature.
Response 10: Line 655 in the article shows t=50 0C - the temperature on the Celsius scale, which is usually denoted by the letter “t”. We do not use absolute temperature, so it is entirely legitimate to use the designation “t”; there is no mistake in our text. (Due to the reviever’s comments, we made some changes.
Now line 655 became line 746).
Comments 11: Line 759: Weird sentence. Suggestion “One can see from Figure 40 that surface modification with solutions of individual silanes leads to the inhibition of copper corrosion, as observed in 8 days of accelerated corrosion tests in a salt spray chamber”.
Response 11: The phrase “solutions of individual silanes” means that we are talking about solutions of individual compounds (not mixtures).
We have rephrased this sentence and replaced the sentence in the text of the article, divided by the respected reviewer, with the corrected one:
«From Figure 40 it can be seen that modification of the copper surface with solutions of individual silanes leads to inhibition of corrosion, as evidenced by the results of 8-day accelerated corrosion tests in a salt spray chamber». (Due to the reviever’s comments, we made some changes. Now line 759 became line 855).
Notes:
Due to the reviever’s comments, we made some changes:
Now Figure 40 became Figure 31.
Comments 12: Line 853: Sentence uncomplete.
Response 12: The sentence on line 853 is the caption to Figure 49. Indeed, the word "atmosphere” is missing at the end of the first sentence. The final version will be:
"Results of outdoor corrosion tests of copper modified with organosilanes in a coastal moderately cold atmosphere”.
We have made a correction to the text of the article. (Due to the reviever’s comments, we made some changes. Now line 853 became line 958).
Notes:
Due to the reviever’s comments, we made some changes:
Now Figure 49 became Figure 40.
Comments 13: Line 879: Typo ~1 m
Response 13: Indeed, there is a typo on line 879 in the first output. Please accept our apologies. The brackets should say either 1μm or 1 micron. We have corrected this typo in the manuscript.
(Due to the reviever’s comments, we made some changes. Now line 879 became line 985).
Round 2
Reviewer 1 Report
Comments and Suggestions for Authors
1. Patina is a green or brown film on the surface of copper produced as a result of oxidation/corrosion of surface.
2. In my opinion its good to have a understanding of surface behavior (hydrophobicity) to establish relationship of surface with corrosion. here are few references:
1. Wang RG, Kaneko J. Hydrophobicity and corrosion resistance of steels coated with PFDS film. Surface Engineering. 2013;29(4):255-263.
2. Peng Wang, Ri Qiu, Dun Zhang, Zhifeng Lin, Baorong Hou, Fabricated super-hydrophobic film with potentiostatic electrolysis method on copper for corrosion protection, Electrochimica Acta, 56(1),2010, 517-522.
Reviewer 2 Report
Comments and Suggestions for Authors
The authors have revised the manuscript satisfactorily and it is recommended for acceptance in its current form.
Author Response
Dear Mr. (Mrs.) Reviewer,
We (the authors) would like to thank you for taking the time to review our article and for your valuable comments and suggestions. Thank you!
Kind regards,
the authors:
Maxim Petrunin, Tatyana Yurasova, Alevtina Rybkina and Liudmila Maksaeva
Reviewer 3 Report
Comments and Suggestions for Authors
Many thanks to the authors for their detailed feedback. Many points have been resolved, but there are still the following points that should be corrected/improved:
- The literature summary is now more specific to corrosion protection for copper and organosilanes as protective surface layers. However, the manuscript still lacks a statement on what this work contributes to advance the state of the art and why the advancements are important. The response letter states that no additives are the novelty, but there is no information in the manuscript.
- In relation to the organosilane work cited, it is explained what was studied, but there is no information on the main findings to support the rationale for this work.
- The methodology description is still difficult to read and it seems that the authors did not want to change anything here. In the end, the editor will decide whether it is acceptable.
Comments on the Quality of English Language
no comments
